# Patch-Wise Random and Noisy CutMix for Privacy-Preserving Split Learning with Vision Transformer

## Abstract

In computer vision, the vision transformer (ViT) has increasingly superseded the convolutional neural network (CNN) for improved accuracy and robustness. Since ViT often comes with large model sizes and high sample complexity, split learning (SL) is a promising solution to training ViT using large memory and computing resources at a server with the sheer amount of private data owned by users or clients. In SL, a ViT is split into two parts under a server-client architecture. The sever stores its upper segment that is associated with multiple clients each of which stores the lower segment. At the cut layer between the upper and lower segments, SL exchanges the cut-layer hidden activations in the forward propagation (FP), referred to as smashed data, and the cut-layer gradients in the backpropagation (BP), which are exposed to various attacks on private training data. To mitigate the risk of data breaches in classification tasks, inspired from the CutMix regularization, we propose a novel privacy-preserving SL framework that injects Gaussian noise into smashed data and mixes randomly chosen patches of smashed data across clients, coined *DP-CutMixSL*. By analysis, we prove that DP-CutMixSL is a differentially private (DP) mechanism amplifying the privacy budget with respect to membership inference attacks in FP. By simulation, we additionally show that DP-CutMixSL protects privacy from reconstruction attacks in FP and from label inference attacks in BP. Surprisingly, DP-CutMixSL even improves accuracy and robustness to imbalanced data distributions over clients, due to the regularization effect of its patch-wise random CutMix operations.

## 1 Introduction

Transformer architecture has originally been developed in the domain of natural language processing (NLP) Vaswani et al. (2017), and its application has recently been extended to various domains including speech recognition Karita et al. (2019) and computer vision (CV) Dosovitskiy et al. (2020b). In particular, the *vision transformer (ViT)* has recently been the new standard architecture in CV, succeeded to the convolutional neural network (CNN) architecture. ViT operations are summarized in two steps: 1) the first step to dividing image data into multiple image *patches*, and 2) the second step to learning the relationship between the patches under the encoder of the transformer. The latter step, dubbed the *self-attention* mechanism, helps to achieve high performance on large datasets, but causes performance degradation on small datasets due to its weak inductive bias. Hence, securing large-scale datasets in ViT is essential yet challenging, especially in distributed learning scenarios where huge data is dispersed to multiple *clients* with limited computing capability Khan et al. (2021); Han et al. (2022). Federated learning (FL) is a promising solution in terms of enjoying these scattered data and computing resources Li et al. (2020); Kairouz et al. (2021). In FL, each client trains a local model to be uploaded to the *server* with their own dataset, while the server yields the global model by taking the weighted average of the local models, leading to data diversity gain without direct data exchange. Such model averaging, however, makes FL not suitable for ViT, which is computationally expensive and often has a large model size.

To cope with this, *split learning (SL)* can be an alternative solution in a sense that the client and server divide the entire model into two fractions, a *lower model segment* and an *upper model segment*, and store each one Gupta and Raskar (2018); Vepakomma et al. (2018). Under this model-split architecture, clients

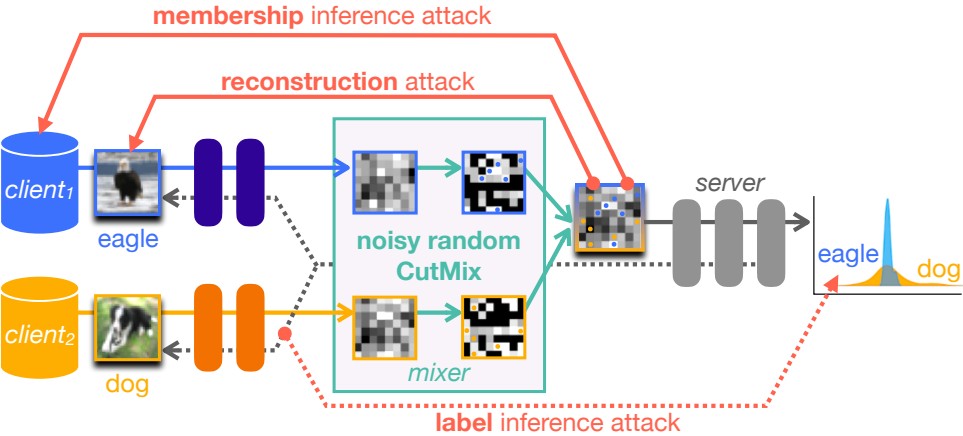

Figure 1: A schematic illustration of DP-CutMixSL with 2 clients.

upload a cut-layer representation, also referred to as *smashed data*, from the SL's forward propagation (FP) and download its corresponding gradient from the SL's backward propagation (BP). However, unlike CNN's smashed data, the smashed data of ViT is not actually "smashed" due to the absence of such a pooling layer, questioning SL's data privacy guarantee.

To this end, we propose a novel distributed learning framework for ViT, coined *DP-CutMixSL*, that is differentially private (DP) under the parallel SL (PSL) architecture. Influenced by CutMix regularization Yun et al. (2019), each client of DP-CutMixSL uploads its own masked smashed data, whose masks are mutually exclusive collectively exhaustive (MECE) in patch-wise, with additive Gaussian noise. Precisely, a novel entity, the *mixer*, generates the two MECE and binary masks with a dirichlet-multinomial distribution Bishop et al. (2007), which are distributed to each of the two clients. Here, the mixer aims to operate while maintaining data privacy to the server, and is implemented through homomorphic encryption Rivest et al. (1978); Pereteanu et al. (2022) or analog communication Koda et al. (2021). Then, each client injects noise following a Gaussian mechanism Abadi et al. (2016); Gil et al. (2013); Girgis et al. (2021) into the smashed data, and returns only a fraction of it to the mixer based on the mask. Next, the mixer combines the aggregated data to yield *DP-CutMix smashed data*, followed by the remaining operations of PSL including server-side FP-BP.

In doing so, DP-CutMixSL is beneficial in terms of robustness against various privacy attacks shown in figure 1. Regarding privacy attacks in SL, membership inference attack Shokri et al. (2017); Rahman et al. (2018) and model inversion attack or reconstruction attack He et al. (2019) in FP try to determine whether or not data is used and restore input data, respectively, while label inference attack Li et al. (2021); Yang et al. (2022) in BP infers the label through the gradient of the cut-layer. Thanks to its randomly punched smashed data as well as Gaussian noise, DP-CutMixSL can provide a strong privacy guarantee under these privacy attacks. Meanwhile, DP-CutMixSL shows improved performance in accuracy and resilience to unbalanced data distributions, thanks to its inherent regularization effect.

**Contributions.** The key contributions of this article are summarized as follows[1]:

- Inspired by CutMix, we propose a new SL-based architecture named DP-CutMixSL aiming to improve privacy guarantee of ViT. In this process, we introduce an entity called mixer on a network consisting of client-server, and define its operations such as pseudorandom sequence generation for yielding DP-CutMix smashed data.

- We theoretically derive the privacy guarantee of DP-CutMixSL against membership inference attacks through DP analysis and experimentally demonstrate it.

---

[1]This work is an extended version of both our previous workshop papers Baek et al. (2022); Oh et al. (2022a), with the addition of extensive experiments involving label inference attacks and an analysis of the subsampled mechanism.

- Through experiments, we verify that DP-CutMixSL is robust against reconstruction attack and label inference attack.

- In addition, we show that DP-CutMixSL outperforms baselines such as PSL and split federated learning (SFL) in terms of accuracy, even for imbalanced data distributions via numerical evaluation.

## 2 Related Works

**Vision Transformers.** The transformer architecture is first used in the NLP field Vaswani et al. (2017), where its core operation is rooted on self-attention mechanism as well as encoder structure with multi-layer perceptron (MLP) and residual connection. In NLP, such transformer-based architecture is extended from Bidirectional Encoder Representations from Transformers (BERT) Devlin et al. (2018), Generative Pre-trained Transformer (GPT) Radford et al. (2018) to GPT-2 Radford et al. (2019), GPT-3 Brown et al. (2020). This paradigm shift from CNN to transformer has reached out to the CV field. ViT, proposed in Dosovitskiy et al. (2020a), is the first of its kind to apply the transformer architecture to the CV field. ViT transforms an input image into a series of image patches, just as a transformer embeds words in text, and learns relationships between image patches, thereby a large-scale dataset is indispensible. This ViT operation enables to extract global spatial information, leading to its robustness against information loss such as patch drop and image shuffling compared to CNN Naseer et al. (2021).

If most of the ViT works are based on the centralized method Carion et al. (2020); Zheng et al. (2021); Chen et al. (2021), several studies have conducted research on distributed implementation of transformer or ViT Hong et al. (2021); Park et al. (2021b); Qu et al. (2021). Hong et al. (2021) has designed FL-based transformer structure targeting text to speech task, while Qu et al. (2021) has explored the performance of FL in ViT when data are heterogeneous. To diagnose COVID-19, Park et al. (2021b) proposed SL-based architecture in ViT, benefiting from its robustness on task-agnostic training.

**Federated & Split Learning.** The key element of the distributed learning framework is to utilize raw data and computing resources spread across the sheer amount of Internet-of-Things (IoT) devices or clients. As the first kind of this, FL enables to acquire data diversity gain through exchanging model parameters Li et al. (2020); Kairouz et al. (2021). FL's model parameter aggregation does not induce data privacy leakage, and what is more, it ensures scalability in terms of increasing accuracy with the number of participating clients Konečný et al. (2015); Park et al. (2021a). Nevertheless, FL has a trouble in running a large-size model, constrained by its limited client-side computation and communication resources, highlighting the need for alternative solutions Konečný et al. (2016); Singh et al. (2019). To this end, SL has appeared as a enabler for large model operation by splitting the entire model into two partitions Gupta and Raskar (2018); Vepakomma et al. (2018); Gao et al. (2020). The initial implementation of SL, which is based on sequential method, used to result in large latency especially with many clients, giving rise to the research on PSL free from this problem. SFL, a combination of FL and SL, is the first form of PSL, allowing simultaneous access by multiple clients Thapa et al. (2020a;b); Gao et al. (2021). One step further, Pal et al. (2021) and Oh et al. (2022b) try to address the low accuracy, communication efficiency, and scalability of SFL.

**Privacy Attacks & Differential Privacy.** As machine learning develops rapidly, several types of privacy attacks have emerged whose goal is to extract information about training data, labels or the model itself. In particular, regarding the privacy attack on distributed learning, Nasr et al. (2018) shows the membership inference attack of an adversary with some auxiliary information on the training data. He et al. (2019) investigates the reconstruction attack occurring on the inference phase of vanilla SL under white-box and black-box settings, while Oh et al. (2022b) measures it emprically on the PSL structure. In addition, for label inference attacks in vanilla SL, Li et al. (2021) handles norm-based and direction-based attacks under black-box setting, and Yang et al. (2022) deals with white-box attacks and GradPerturb as a solution for them.

Accordingly, many studies have been conducted to protect information from various privacy attacks. One line of works first introduced the application of DP analysis technique to deep learning models Dwork (2008); Abadi et al. (2016). PixelDP, designed for SFL, is proposed as a DP-based defence to adversarial examples, providing certified robustness to AI/ML models, while Wu et al. (2022) applies the concept of DP to FL.

Meanwhile, Reńyi DP (RDP) is presented to facilitate the composition between heterogeneous mechanisms while providing tight bounds with fewer computations Mironov (2017). Such DP or RDP bounds can be tighter through Mixup Zhang et al. (2017); Verma et al. (2019) with the help of its inherent distortion property Koda et al. (2021); Borgnia et al. (2021); Lee et al. (2019). Another privacy amplification is possible via subsampling Balle et al. (2018) as well as shuffling Erlingsson et al. (2019).

## 3 DP-CutMixSL: Split Learning With Random CutMix for ViT

In this section, we demonstrate a proposed *differentially private CutMixSL (DP-CutMixSL)* framework, aiming to improve ViT's privacy guarantee while still ensuring its accuracy. Consider a network with a set of clients $\mathbb{C} = \{1, 2, \cdots, n\}$ and a single server. Here, let $i$ be the subscript for the client. The dataset of the $i$-th client is expressed as $\mathbb{D}_i$, consisting of multiple tuples of input data $\mathbf{x}_i$ and its one-hot encoded ground-truth label $\mathbf{y}_i$. We denote the $i$-th entire network as $\mathbf{w}_i = [\mathbf{w}_{c,i}, \mathbf{w}_s]^{\mathrm{T}}$, where $\mathbf{w}_{c,i}$, $\mathbf{w}_s$, and $(\cdot)^{\mathrm{T}}$ represent the $i$-th lower model segment, the upper model segment, and the transpose function, respectively. In addition, $f_i$ and $g$ indicate mapping from input data to smashed data via $\mathbf{w}_{c,i}$ and mapping from smashed data to output via $\mathbf{w}_s$, while $P$ means patch size.

### A   Baseline: Parallel Split Learning

This subsection first revisits the PSL operation on ViT. For the sake of convenience, we assume that the cut-layer is located between the embedding layer and the transformer in the ViT structure, so that the client converts the input image into *embedded patches* through the lower model segment, followed by the server forward and back propagates it through the upper model segment.

**Patch Embedding.**    At the $i$-th client, the input data $\mathbf{x}_i \in \mathbb{R}^{H \times W \times C}$ is first divided into $N$ image patches, so that $\mathbf{x}_i$ is transformed into $\mathbf{x}'_i \in \mathbb{R}^{N \times P^2 \times C}$, where the number of image patches $N$ equals to $\frac{H \cdot W}{P^2}$. Then, each image patch is linearly projected, and a class token is attached in front of image patches. Finally, an embedding vector is added for positional embedding, producing embedded patches $\mathbf{s}_i = [\mathbf{s}_{i,0}, \mathbf{s}_{i,1}, \cdots, \mathbf{s}_{i,N}]$ corresponding to the smashed data on ViT with PSL. This completes the 1D embedding step that generates the transformer's input. Note that the positional embedding is trainable and parameterizable with $\mathbf{w}_{c,i}$. We abbreviate the smashed data obtained through the entire 1D embedding process above as follows:

$$\mathbf{s}_i = f_i(\mathbf{x}_i). \tag{1}$$

**FP & BP in Transformer.**    At the server, $\mathbf{s}_i$ for all $i \in \mathbb{C}$ is aggregated and passes through the upper model segment $\mathbf{w}_s$ to yield *softmax output* denoted by $g(\mathbf{s}_i)$. Using the cross entropy function $CE(p,q) = -\sum_j q_j \log p_j$ where $j$ is the subscript for the element, the loss for the $i$-th smashed data can be represented as $L_i = CE(g(\mathbf{s}_i), \mathbf{y}_i)$. If we generalize this to batch $\mathbb{B}_i \subset \mathbb{D}_i$ of size $b$, $L_i$ becomes:

$$L_i = \frac{1}{b} \sum_{(\mathbf{x}_i, \mathbf{y}_i) \in \mathbb{B}_i} CE(g(f_i(\mathbf{x}_i)), \mathbf{y}_i). \tag{2}$$

For all $i$, based on loss $L_i$, the server sends the $i$-th cut-layer gradient indicated by $\nabla_{\mathbf{s}_i} L_i$ to the $i$-th client, enabling weight update in the server and client as in the following formula:

$$\begin{bmatrix} \mathbf{w}_s \\ \mathbf{w}_{c,i} \end{bmatrix} \leftarrow \begin{bmatrix} \mathbf{w}_s \\ \mathbf{w}_{c,i} \end{bmatrix} - \eta \begin{bmatrix} \sum_{i \in \mathbb{C}} d_i \cdot (\nabla_{\mathbf{w}_s} L_i) \\ \nabla_{\mathbf{w}_{c,i}} L_i \end{bmatrix}, \tag{3}$$

where $\eta$ is the learning rate and $d_i = |\mathbb{D}_i| / \sum_i |\mathbb{D}_i|$.

**Motivation: Privacy-Preserving Parallel SL in ViT.**    As organized in figure 2, the above operation of ViT with PSL has the following fundamental characteristics compared to that with CNN, consequently highlighting the need for additional solutions.

*1) Absence of Pooling Layer*: While CNN contains a pooling layer, ViT often skips the pooling layer except in cases like pooling-based ViT (PiT) Heo et al. (2021b). Due to this difference, ViT not only has less distortion

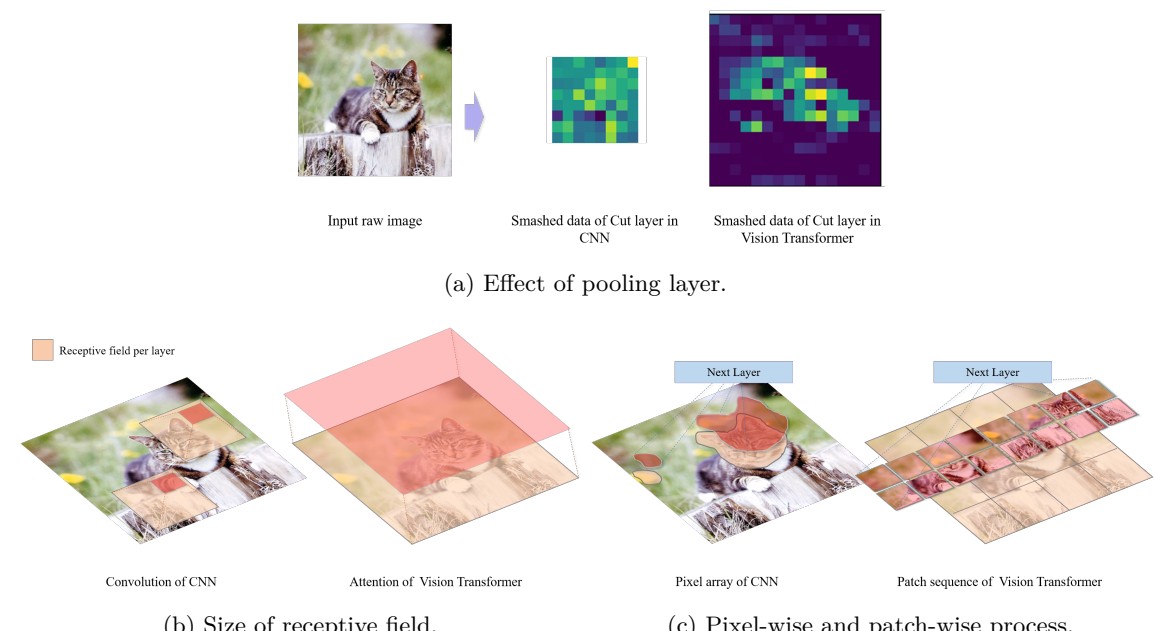

(a) Effect of pooling layer.

(b) Size of receptive field.  (c) Pixel-wise and patch-wise process.

Figure 2: Comparison of CNN and ViT operation from various perspectives.

of the hidden representation, but also its size is the same as that of the input, causing a significant amount of privacy leakage. On the bright side, this makes regularization on hidden representations more fruitful, just like regularization on input data.

*2) Receptive Field Size*: A CNN with a convolutional layer is specialized in catching local spatial information of an image, in other words, its receptive field size is small. Conversely, the receptive field of ViT is large enough to learn global spatial information, with the help of its self-attention mechanism. Because of this, ViT is more suitable for producing generalized models compared to CNN, but large-scale datasets are required to unleash the full potential of ViT due to its low inductive bias Baxter (2000). Data regularization can address this large-scale dataset requirement Steiner et al. (2021). In particular, due to its small receptive field, ViT has robustness against large noise applied to part of the image, such as patch drop or image shuffling Naseer et al. (2021), and is thereby suitable for CutMix regularization.

*3) Patch-Wise Processing*: When CNN processes all operations such as convolution at pixel-wise, ViT performs patch-wise processing such as dividing and embedding images based on a novel unit named patch, which is a square-shaped collection consisting of multiple pixels. Even the self-attention mechanism, which is the core operation of ViT, also operates at the patch-level, and focuses on identifying spatial relations between these patches. As an extension of this patch-wise ViT operation, we deduce that novel patch-wise regularization can be the key to further improving ViT's performance.

Combining *1)-3)* leads to one converged solution, a novel regularization that is a patch-wise variant of CutMix applied to the hidden representations of ViT with PSL. Its CutMix-like properties can maximize the advantages of ViT mentioned in *1)*, while addressing both privacy leakage problem in terms of the client sending fraction, but not entire, of the smashed data. Also, while providing optimized regularization for the receptive field of ViT in *2)*, it can overcome the poor performance of ViT with an insufficient dataset thanks to its inherent regularization effect. Lastly, as suggested in *3)*, the novel patch-wise regularization has the potential to suit ViT operation rather than cutting or replacing a box-shaped region consisting of multiple pixels as in a Vanilla CutMix.

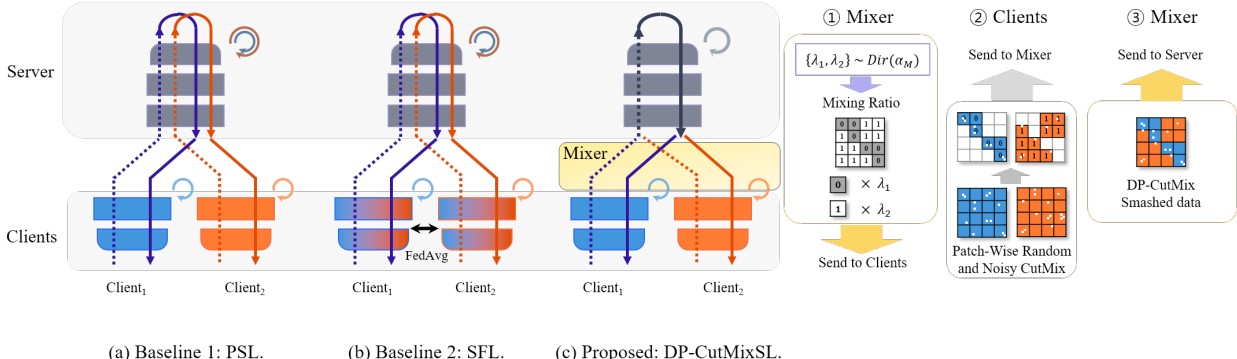

(a) Baseline 1: PSL.          (b) Baseline 2: SFL.          (c) Proposed: DP-CutMixSL.

Figure 3: Illustrations of (a) parallel SL (PSL), (b) split federated learning (SFL), and (c) DP-CutMixSL.

## B    Proposed: Patch-Wise Random and Noisy CutMix for ViT

In this subsection, we propose a *Patch-Wise Random and Noisy CutMix (Random CutMix)*, where the client sends a fraction of the smashed data with Gaussian noise to ensure privacy, as depicted in figure 3. Before explaining the details of Random CutMix's operation, a novel entity called mixer is introduced on the existing network consisting of server and client. While the server is *honest-but-curious* and causes privacy leakage, the mixer does not, so the core operations of Random Cutmix are performed without privacy concerns. For the mixer, the concept of homomorphic encryption can be applied Rivest et al. (1978); Pereteanu et al. (2022). With homomorphic encryption, computation can be processed under privacy preservation, as the client transmits homomorphically encrypted smashed data, and the server subsequently produces encrypted output as a mixer. Alternatively, the mixer can be an analog channel, as in Koda et al. (2021).

**Pseudorandom Binary Sequence Generation in Mixer.**    Let $\mathbb{C}_k$ be a set of client pairs obtained by non-restored extraction of $k$ clients from the client set $\mathbb{C}$, and define its elements as *mixing groups*. For instance, when $\mathbb{C} = \{1, 2, 3, 4\}$, $\mathbb{C}_2 = \{\{1, 2\}, \{3, 4\}\}$. Then, for the $i$-th client in the mixing group $\{1, \cdots, k\}$, the mixer first calculates the $i$-th *mixing ratio* $\lambda_i$ following the symmetric dirichlet distribution Bishop et al. (2007) with *mask distribution* $\alpha_M$. For the $i$-th client, the mixer extracts $\lceil \lambda_i \cdot N \rceil$ elements from the set $\{1, \cdots, N\}$ without replacement to generate a mask $\mathbf{M}_i$, which allocates the total $N$ patches to $k$ clients without overlapping. For tractability we assume $k = 2$ in the following equations.

**Random CutMix.**    After the mixer sends $\mathbf{M}_i$ to the $i$-th client, each $i$-th client injects noise into the smashed data and labels according to the Gaussian mechanism, and then punches smashed data based on its corresponding mask $\mathbf{M}_i$ as follows:

$$\bar{\mathbf{s}}_i = \mathbf{M}_i \odot (\mathbf{s}_i + n_{s,i}), \qquad \bar{\mathbf{y}}_i = \lambda_i \cdot (\mathbf{y}_i + n_{y,i}), \tag{4}$$

where the size of the matrices $n_{s,i}$ and $n_{y,i}$ are equal to those of $\mathbf{s}_i$ and $\mathbf{y}_i$, respectively, and their elements follow a zero-mean Gaussian distribution with variances $\sigma_s^2$ and $\sigma_y^2$. For $\{i, i'\} \in \mathbb{C}_2$, if each client uploads $(\bar{\mathbf{s}}_i, \bar{\mathbf{y}}_i)$, the mixer then yields *DP-CutMix smashed data* with the following Mixup:

$$\tilde{\mathbf{s}}_{\{i,i'\}} = \bar{\mathbf{s}}_i + \bar{\mathbf{s}}_{i'}, \qquad \tilde{\mathbf{y}}_{\{i,i'\}} = \bar{\mathbf{y}}_i + \bar{\mathbf{y}}_{i'}. \tag{5}$$

The mixer uploads this DP-CutMix smashed data and its label to the server, followed by the server propagates it to the upper model segment to produce the loss of the following formula:

$$\tilde{L}_{\{i,i'\}} = \frac{1}{b} \sum CE(g(\tilde{\mathbf{s}}_{\{i,i'\}}), \tilde{\mathbf{y}}_{\{i,i'\}}), \tag{6}$$

which is the counterpart of equation 2.

The operation up to this point is performed in parallel to other mixing groups in $\mathbb{C}_2$. When the mixer receives cut-layer gradient $\nabla_{\tilde{\mathbf{s}}_{\{i,i'\}}} \tilde{L}_{\{i,i'\}}$ from the server in BP, the effect of the FP flow interrupted by mutually

---

**Algorithm 1** Operation of DP-CutMixSL with $k = 2$.

---

**Input:** mask distribution $\alpha_M$

  **/\*Execute in mixer\*/**

  **function 1.** PSEUDORANDOM SEQUENCE GENERATION

    Generate $\mathbb{C}_2$ by repeating extraction without replacement of 2 clients from client set $\mathbb{C}$

    Sample $\{\lambda_1, .., \lambda_k\} \sim \text{Dir}(\alpha_M)$

    Generate binary masks $\{\mathbf{M}_1, .., \mathbf{M}_k\}$ according to $\{\lambda_1, .., \lambda_k\}$

    **Return** $\mathbf{M}_i$ to $i$-th client for all $i \in \mathbb{C}_k$

  **function 2.** RANDOM CUTMIX

    Generate $\{\tilde{\mathbf{s}}_{\{i,i'\}}, \tilde{\mathbf{y}}_{\{i,i'\}}\}$ through equation 5

    **Return** $\{\tilde{\mathbf{s}}_{\{i,i'\}}, \tilde{\mathbf{y}}_{\{i,i'\}}\}$ to server

  **function 3.** CUT-LAYER GRADIENT SPLITTING

    Split $\nabla_{\tilde{\mathbf{s}}_{\{i,i'\}}} \tilde{L}_{\{i,i'\}}$ into $\nabla_{\tilde{\mathbf{s}}_{\{i,i'\}}} \mathbf{M}_i \odot \tilde{L}_{\{i,i'\}}$ and $\nabla_{\tilde{\mathbf{s}}_{\{i,i'\}}} \mathbf{M}_{i'} \odot \tilde{L}_{\{i,i'\}}$ through equation 8

    **Return** $\nabla_{\tilde{\mathbf{s}}_{\{i,i'\}}} \mathbf{M}_{i(i')} \odot \tilde{L}_{\{i,i'\}}$ to $i(i')$-th client

  **while** $\mathbf{w}_i$ not converged **do**

    PSEUDORANDOM SEQUENCE GENERATION

    **/\*Execute in client $i \in \mathbb{C}_k$\*/**

    Generate $\mathbf{s}_i$ through equation 1

    Generate $(\bar{\mathbf{s}}_i, \bar{\mathbf{y}}_i)$ as in equation 4 based on $\mathbf{M}_i$ and Gaussian mechanism

    Send $(\bar{\mathbf{s}}_i, \bar{\mathbf{y}}_i)$ to mixer

    RANDOM CUTMIX

    **/\*Execute in server**

    Generate loss $\tilde{L}_{\{i,i'\}}$ through server-side FP in equation 6

    Update $\mathbf{w}_s$ through equation 9

    Send $\nabla_{\tilde{\mathbf{s}}_{\{i,i'\}}} \tilde{L}_{\{i,i'\}}$ to mixer

    CUT-LAYER GRADIENT SPLITTING

    **/\*Execute in client $i \in \mathbb{C}_k$\*/**

    Update $\mathbf{w}_{c,i}$ through equation 9

  **end while**

---

exclusive masks should be reflected in the gradient of each client before it is sent to them. For this, the mixer divides the cut-layer gradient into the gradient of the $i$-th client and the gradient of the $i'$-th client as follows:

$$\nabla_{\tilde{\mathbf{s}}_{\{i,i'\}}} \tilde{L}_{\{i,i'\}} = \mathbf{M}_i \odot \nabla_{\tilde{\mathbf{s}}_{\{i,i'\}}} \tilde{L}_{\{i,i'\}} + \mathbf{M}_{i'} \odot \nabla_{\tilde{\mathbf{s}}_{\{i,i'\}}} \tilde{L}_{\{i,i'\}} \tag{7}$$

$$= \underbrace{\nabla_{\tilde{\mathbf{s}}_{\{i,i'\}}} (\mathbf{M}_i \odot \tilde{L}_{\{i,i'\}})}_{\text{gradient for client } i} + \underbrace{\nabla_{\tilde{\mathbf{s}}_{\{i,i'\}}} (\mathbf{M}_{i'} \odot \tilde{L}_{\{i,i'\}})}_{\text{gradient for client } i'}. \tag{8}$$

Note that since $\mathbf{M}_{i(i')}$ is a binary matrix, all elements are constants of 0 or 1, and the derivation from equation 7 to equation 8 is trivial by the constant multiply rule of derivate.

After the mixer transmits each gradient to the corresponding client, the weight update of the $i(i')$-th client and server through the BP is available as shown below:

$$\begin{bmatrix} \mathbf{w}_s \\ \mathbf{w}_{c,i(i')} \end{bmatrix} \leftarrow \begin{bmatrix} \mathbf{w}_s \\ \mathbf{w}_{c,i(i')} \end{bmatrix} - \eta \begin{bmatrix} \sum_{\{i,i'\} \in \mathbb{C}_2} (d_i + d_{i'}) \cdot (\nabla_{\mathbf{w}_s} \tilde{L}_{\{i,i'\}}) \\ \nabla_{\mathbf{w}_{c,i(i')}} \mathbf{M}_{i(i')} \odot \tilde{L}_{\{i,i'\}} \end{bmatrix}. \tag{9}$$

This completes DP-CutMixSL's single communication round. The above formulas for DP-CutMixSL assume $k = 2$, but the generalization for $k > 2$ is trivially extensible. Also, as detailed in appendix A, the proposed Random CutMix includes some observations about the accuracy with respect to hyperparameters. Following this, hereafter a Random CutMix in which $k$ and $\alpha_M$ are both equal to 2 is used without further explanation.

In short, in this section, we first review the SL framework with ViT, and then propose DP-CutMixSL, which combines it with the novel regularization scheme, coined Random CutMix, optimized for the characteristics

of ViT. The detailed operation of the DP-CutMixSL aided by the core functions of mixer is summarized in algorithm 1. While controlling the mixing group size only in this work, it is possible to increase the number of FP flows by combinatorially setting the mixing group several times during single FP as in Oh et al. (2022b), focusing on its augmentation properties. This can be a solution to the low inductive bias of ViT, which is deferred to future research. In addition, as in Oh et al. (2022a), under the implicit premise of sharing the mask using the seed concept, mixer's functions can be distributed to clients and server, allowing asynchronous training of DP-CutMixSL.

## 4 Differential Privacy Analysis on Smashed Data

In this section, we theoretically analyze the differential privacy (DP) bound of DP-CutMixSL and validate its effectiveness for privacy guarantees. Unlike existing DP works that focus on the privacy guarantees of samples and labels, we conduct DP analysis from the perspective of smashed data, which is highly correlated with the sample particularly in ViT, and its labels. This is in the same context as analyzing the privacy leakage of model parameters or gradient in FL.

Given the ViT's patch size of $P$, a dataset $\mathcal{D} = \{(\mathbf{s}_1, \mathbf{y}_1), .., (\mathbf{s}_n, \mathbf{y}_n)\}$ consists of $n$ clients' pairs of smashed data $\mathbf{s}_i \in \mathbb{R}^{P^2 \times N \times C} = \mathbb{R}^{D_s}$ and the corresponding label $\mathbf{y}_i \in \mathbb{R}^L = \mathbb{R}^{D_y}$ is a one-hot vector of size $L$, where $N$ and $C$ denote the number of patches and channels, respectively. For the sake of convenience, we assume that $\mathbf{s}_i$ and $\mathbf{y}_i$ are normalized so that $\mathbf{s}_i \in [0, \Delta]^{D_s}$ and $\mathbf{y}_i \in [0, 1]^{D_y}$, respectively. Before going further, the definition of Central DP (CDP) Dwork et al. (2006) is organized as follows:

**Definition 1** (($\varepsilon, \delta$)-CDP)**.** *For $\varepsilon \geq 0$ and $\delta > 0$, we say that a randomized mechanism $\mathcal{M} : \mathcal{D} \to \mathcal{R}$ is $(\varepsilon, \delta)$-CDP if it satisfies the following inequality for any adjacent $D, D' \in \mathcal{D}$ and $U \subset \mathcal{R}$:*

$$Pr[\mathcal{M}(D) \in U] \leq e^\varepsilon \cdot Pr[\mathcal{M}(D')] \in U] + \delta, \tag{10}$$

At this point, a small $\varepsilon$ indicates a high privacy level implying that one cannot distinguish whether $D$ or $D'$ is used to produce an outcome of mechanism. Although CDP is widely used when analyzing Gaussian mechanisms, we also use the Reńyi DP (RDP) Mironov (2017), defined below, given the tractable interpretation of its composition rule:

**Definition 2** (($\alpha, \epsilon$)-RDP)**.** *A randomized mechanism $\mathcal{M} : \mathcal{D} \to \mathcal{R}$ is said to have $\epsilon$-RDP of order $\alpha$, or $(\alpha, \epsilon)$-RDP for short, if for any adjacent $D, D' \in \mathcal{D}$ it holds that:*

$$D_\alpha(\mathcal{M}(D) \| \mathcal{M}(D')) \leq \epsilon. \tag{11}$$

In addition, Mironov (2017) proves that every RDP mechanism is also $(\varepsilon, \delta)$-CDP. Especially in Mironov (2017), when the mechanism is $(\alpha, \epsilon)$-RDP, then it is $(\epsilon + \frac{\log(1/\delta)}{\alpha - 1}, \delta)$-DP for $0 < \delta < 1$. For convenience, this section derives the RDP guarantee and the simulations in the next section measure the CDP guarantee. Furthermore, for tractability, this section assumes that there is a single uploaded smashed data and label for each client, so subscript $i$ displays the client and its smashed data and label simultaneously.

### A   RDP Bound of DP-SL

First, we can consider a simple DP mechanism, denoted $\mathcal{M}_1$, where smashed data and labels are injected with noise following a Gaussian mechanism, respectively. Then, the outputs of $\mathcal{M}_1$ (DP-SL) are as follows:

$$\bar{\mathbf{s}}_i = \mathbf{s}_i + n_{s,i}, \qquad \bar{\mathbf{y}}_i = \mathbf{y}_i + n_{y,i}, \tag{12}$$

where $n_{s,i} \sim \mathcal{N}(0, \sigma_s^2 I_{D_s})$ and $n_{y,i} \sim \mathcal{N}(0, \sigma_y^2 I_{D_y})$ for all $i$ and some $(\sigma_s, \sigma_y)$.

We now demonstrate the RDP guarantee of DP-SL as follows:

**Proposition 1.** *For all integer $\alpha \geq 2$, $\mathcal{M}_1$ is $(\alpha, \epsilon_1(\alpha))$-RDP, where*

$$\epsilon_1(\alpha) = \frac{\alpha \Delta^2 D_s}{2\sigma_s^2} + \frac{\alpha D_y}{2\sigma_y^2}. \tag{13}$$

Proof. *Starting from the definition 2 and the Rényi divergence formula with multi-variate Gaussian distributions Gil et al. (2013), the RDP bound of $\mathcal{M}_1$, denoted by $\epsilon_1(\alpha)$, can be expressed as:*

$$\epsilon_1(\alpha) = \sup_{D,D'} D_\alpha(\mathcal{M}_1(D)||\mathcal{M}_1(D')) = \sup_{D,D'} \underbrace{\frac{\alpha \cdot ||\mu_s^D - \mu_s^{D'}||^2}{2\sigma_s^2}}_{=\epsilon_{1,s}(\alpha)} + \underbrace{\frac{\alpha \cdot ||\mu_y^D - \mu_y^{D'}||^2}{2\sigma_y^2}}_{=\epsilon_{1,y}(\alpha)}, \tag{14}$$

*since $\bar{\mathbf{s}}_i \sim \mathcal{N}(\mu_s^D, \sigma_s^2 I_{D_s})$ and $\bar{\mathbf{y}}_i \sim \mathcal{N}(\mu_y^D, \sigma_y^2 I_{D_y})$, where $\mu_s^D$ and $\mu_y^D$ indicate the average of smashed data and that of label, belonging to dataset $D$. It is noteworthy that $\epsilon_1(\alpha)$ is represented as the sum of RDP bound for smashed data $\epsilon_{1,s}(\alpha)$ and RDP bound for label $\epsilon_{1,y}(\alpha)$ via the sequential composition rule, aiming to induce smashed data-label pairwise RDP bound.*

*Here, by using assumptions about the pixel-wise upper bound of the smashed data and labels ($\mathbf{s}_i \in [0,\Delta]^{D_s}$ and $\mathbf{y}_i \in [0,1]^{D_y}$), we have:*

$$||\mu_s^D - \mu_s^{D'}||^2 \leq \Delta^2 \cdot D_s, \qquad ||\mu_y^D - \mu_y^{D'}||^2 \leq 1^2 \cdot D_y = D_y. \tag{15}$$

*Combining equation 14 and equation 15 concludes the proof.* ∎

## B  RDP Bound of DP-MixSL

Additionally, we can think of a DP mechanism called $\mathcal{M}_2$, which takes a Mixup between $n$ smashed data and labels, followed by a Gaussian DP mechanism $\mathcal{M}_1$ Lee et al. (2019). For the mixing ratio $\lambda_i$ of the $i$-th smashed data and label, the output of $\mathcal{M}_2$ (DP-MixSL) is expressed as follows:

$$\hat{\mathbf{s}} = \sum_{i=1}^{n} (\lambda_i \cdot \mathbf{s}_i) + \hat{n}_s, \qquad \hat{\mathbf{y}} = \sum_{i=1}^{n} (\lambda_i \cdot \mathbf{y}_i) + \hat{n}_y, \tag{16}$$

where $\hat{n}_s$ and $\hat{n}_y$ follow $\mathcal{N}(0, \sigma_s^2 I_{D_s})$ and $\mathcal{N}(0, \sigma_y^2 I_{D_y})$ respectively, and $\sum_{i=1}^{n} \lambda_i = 1$.

We present the privacy guarantee of DP-MixSL as belows:

**Proposition 2.** *For all integer $\alpha \geq 2$, $\mathcal{M}_2$ is $(\alpha, \epsilon_2(\alpha))$-RDP, where*

$$\epsilon_2(\alpha) = (\max_{i\in\mathbb{C}} \lambda_i)^2 \left(\frac{\alpha\Delta^2 D_s}{2\sigma_s^2} + \frac{\alpha D_y}{2\sigma_y^2}\right). \tag{17}$$

Proof. *Consider the output of DP-MixSL where n smashed data and labels are mixed up, and their pixel-wise upper bound and dimension. Then, for two adjacent datasets $D$ and $D'$ (i.e. only $i'$-th elements are different, $1 \leq i' \leq n$), we have:*

$$||\mu_s^D - \mu_s^{D'}||^2 \leq (\lambda_{i'}\Delta)^2 D_s, \qquad ||\mu_y^D - \mu_y^{D'}||^2 \leq \lambda_{i'}^2 D_y. \tag{18}$$

*Here, equation 18 is maximized when $\lambda_{i'}$ is the maximum value of $\lambda_i$ for all $i$. Expressing this is as follows:*

$$(\lambda_{i'}\Delta)^2 D_s \leq (\max_{i\in\mathbb{C}} \lambda_i \cdot \Delta)^2 D_s, \qquad \lambda_{i'}^2 D_y \leq (\max_{i\in\mathbb{C}} \lambda_i)^2 D_y. \tag{19}$$

*Recalling the Rényi divergence formula and combining it with equation 19 completes the proof.* ∎

## C  RDP Bound of DP-CutMixSL

Note that the CutMix behavior is equivalent to the Mixup behavior after Cutout. Then, the DP mechanism $\mathcal{M}_3$ for the proposed DP-CutMixSL first performs a Patch-Wise Random Cutout on the smashed data and its label, and then proceeds with $\mathcal{M}_2$. Here, the outputs of $\mathcal{M}_3$, DP-CutMix smashed data and its label, are as follows:

$$\tilde{\mathbf{s}} = \sum_{i=1}^{n} (\mathbf{M}_i \odot \mathbf{s}_i) + \tilde{n}_s, \qquad \tilde{\mathbf{y}} = \sum_{i=1}^{n} (\lambda_i \cdot \mathbf{y}_i) + \tilde{n}_y, \tag{20}$$

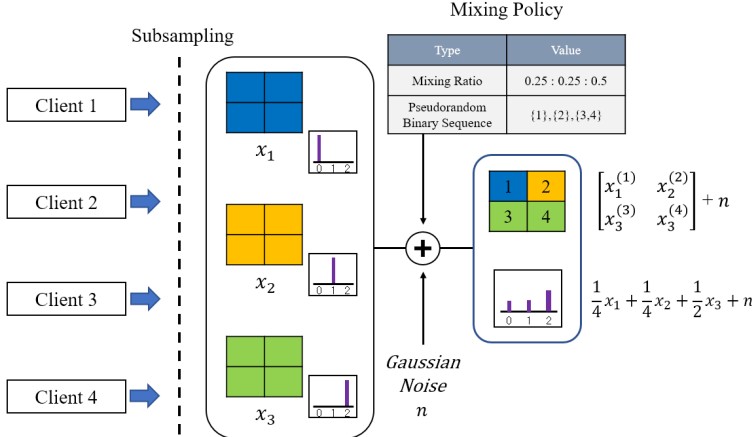

Figure 4: An illustration of DP-CutMixSL with subsampling when $n = 4$ and $k = 3$.

where $\mathbf{M}_i$ are MECE binary masks with $N_i = \lceil \lambda_i \cdot N \rceil$ number of non-zero patches for all $i$, $\tilde{n}_s \sim \mathcal{N}(0, \sigma_s^2 I_{D_s})$, and $\tilde{n}_y \sim \mathcal{N}(0, \sigma_y^2 I_{D_y})$.

Then, the following privacy guarantee of DP-CutMixSL is induced:

**Proposition 3.** *For all integer $\alpha \geq 2$, $\mathcal{M}_3$ is $(\delta, \epsilon_3(\alpha))$-RDP, where*

$$\epsilon_3(\alpha) = (\max_{i \in \mathbb{C}} \lambda_i) \cdot \left( \frac{\alpha \Delta^2 D_s}{2\sigma_s^2} + (\max_{i \in \mathbb{C}} \lambda_i) \frac{\alpha D_y}{2\sigma_y^2} \right). \tag{21}$$

Proof. *If we consider the mean of $\tilde{\mathbf{s}}$ for two adjacent datasets $D$ and $D'$, where only the $i'$-th element is different, $N_{i'} P^2 C$ pixels among the total $D_s$ pixels are different and the rest are identical. At this time, considering the upper bound of the pixel-level, the following inequality is established:*

$$||\mu_s^D - \mu_s^{D'}||^2 \leq (N_{i'} P^2 C) \Delta^2 = (\lambda_{i'} N P^2 C) \Delta^2 = \lambda_{i'} \Delta^2 D_s. \tag{22}$$

*When $\lambda_{i'} = \max_{i \in \mathbb{C}} \lambda_i$, that is to say, $N_{i'} = \max_{i \in \mathbb{C}} N_i$, equation 22 is maximized as follows:*

$$||\mu_s^D - \mu_s^{D'}||^2 \leq (\max_{i \in \mathbb{C}} \lambda_i) \Delta^2 D_s \tag{23}$$

$$= (\max_{i \in \mathbb{C}} N_i) P^2 C \Delta^2. \tag{24}$$

*Recall the Rényi divergence formula, and substitute equation 24 to obtain a privacy guarantee for DP-CutMix smashed data. Since $\mathcal{M}_3$ is identical to $\mathcal{M}_2$ in terms of labels, the proof is completed by applying this to the RDP sequential composition rule together with $\mathcal{M}_2$'s label privacy guarantee.* ∎

It's worth noting that the operations of $\mathcal{M}_3$ and $\mathcal{M}_2$ are equivalent in terms of labels.

## D    Main Result

We can compare the RDP bound of DP-CutMixSL to those of DP-SL and DP-MixSL as follows:

**Theorem 1.** *For a given order $\alpha \geq 2$, the RDP privacy budgets $\epsilon_1(\alpha)$, $\epsilon_2(\alpha)$, and $\epsilon_3(\alpha)$ of DP-SL, DP-MixSL and DP-CutMixSL satisfy the following inequality:*

$$\epsilon_2(\alpha) \leq \epsilon_3(\alpha) \leq \epsilon_1(\alpha), \tag{25}$$

*where*

$$\epsilon_1(\alpha) = \epsilon_{1,s}(\alpha) + \epsilon_{1,y}(\alpha), \tag{26}$$

$$\epsilon_2(\alpha) = \lambda_{max}^2 (\epsilon_{1,s}(\alpha) + \epsilon_{1,y}(\alpha)), \tag{27}$$

$$\epsilon_3(\alpha) = \lambda_{max}(\epsilon_{1,s}(\alpha) + \lambda_{max} \cdot \epsilon_{1,y}(\alpha)), \tag{28}$$

in which $\epsilon_{1,s}(\alpha) = \frac{\alpha \Delta^2 D_s}{2\sigma_s^2}$, $\epsilon_{1,y}(\alpha) = \frac{\alpha D_y}{2\sigma_y^2}$, and $\lambda_{max} = \max_{i \in \mathbb{C}} \lambda_i$.

Proof. *Combining propositions 1 through 3 completes the proof.* ∎

Theorem 1 provides the following 4 observations about DP-CutMixSL.

**Privacy-Accuracy Trade-Off ($k = n$).** There are 2 major *privacy-accuracy trade-off* with theorem 1. On the one hand, the RDP guarantee of DP-CutMixSL is improved compared to that of DP-SL, but is weaker than that of DP-MixSL, whereas DP-CutMixSL outperforms DP-MixSL in terms of accuracy as will be shown in section 5. In DP-MixSL, a large number of samples are superposed in the entire image (i.e. superposition), but privacy leakage is relatively low because they are blurred, whereas in DP-CutMixSL, although privacy leakage occurs only in a fraction of the image (i.e. masking), the sample is leaked as it is, resulting in a performance gap. Counterintuitively, the accuracy of DP-CutMixSL becomes higher than that of DP-MixSL.

On the other hand, regarding $\lambda_{max} \in [1/n, 1]$, when $\lambda_{max}$ reaches $1/n$ (i.e. $|\mathbb{C}|$ increases or $\alpha_M$ goes to $\infty$), the privacy guarantee is maximized and the equality constraint for theorem 1's inequalities is satisfied, but the accuracy decreases as in figure 10b and figure 10c, leading to another privacy-accuracy trade-off.

**RDP-CDP Conversion.** We can measure the CDP guarantee of DP-CutMixSL by applying the aforementioned RDP-to-CDP conversion to theorem 1, which is based on the RDP guarantee. Additionally, the effect of the mixing group size can be reflected by using the CDP bound formula of the subsampled mechanism in Wang et al. (2019), whereas theorem 1 implicitly assumes that the mixing group size is equal to $n$. Thereby, for $k < n$, we can derive the CDP guarantee of DP-CutMixSL with subsampling ($\mathcal{M}^3 \circ$ **subsample**) as below, whose key operation consists of 1) a subsampling mechanism that randomly selects $k$ out of a total of $n$ datapoints (reflecting the mixing group size), and 2) operation of $\mathcal{M}^3$ as depicted in figure 4.

**Corollary 1.** *For all integer $\alpha \geq 2$ and $0 < \delta < 1$, the DP privacy budgets $\varepsilon_1'(\delta)$, $\varepsilon_2'(\delta)$, and $\varepsilon_3'(\delta)$ of $\mathcal{M}^1 \circ$ **subsample**, $\mathcal{M}^2 \circ$ **subsample** and $\mathcal{M}^3 \circ$ **subsample** satisfy the following inequality:*

$$\varepsilon_2'(\delta) \leq \varepsilon_3'(\delta) \leq \varepsilon_1'(\delta), \tag{29}$$

*where*

$$\varepsilon_1'(\delta) = \log\left(1 + \frac{k}{n}(e^{\epsilon_1(\alpha) + \varepsilon_o(\delta)} - 1)\right), \tag{30}$$

$$\varepsilon_2'(\delta) = \log\left(1 + \frac{k}{n}(e^{\epsilon_2(\alpha) + \varepsilon_o(\delta)} - 1)\right), \tag{31}$$

$$\varepsilon_3'(\delta) = \log\left(1 + \frac{k}{n}(e^{\epsilon_3(\alpha) + \varepsilon_o(\delta)} - 1)\right), \tag{32}$$

*in which $k_2^* = \sqrt{\frac{\epsilon_{1,s}(\alpha) + \epsilon_{1,y}(\alpha)}{\varepsilon_o(\delta)}}$ and $k_3^* = \sqrt{\frac{\epsilon_{1,y}(\alpha)}{\varepsilon_o(\delta)}}$ minimize $\varepsilon_2'(\delta)$ and $\varepsilon_3'(\delta)$ under the assumption that $\lambda_i = 1/k \; \forall i$, respectively, and $\varepsilon_o(\delta) = \frac{\log(1/\delta)}{\alpha - 1}$.*

Sketch of Proof. *Recall theorem 1, and apply it to the DP bound formula of the subsampled mechanism of Wang et al. (2019) (if $\mathcal{M}$ is $(\varepsilon, \delta)$-DP, then the subsampled mechanism $\mathcal{M} \circ$ **subsample** is $(log(1 + \gamma(e^\varepsilon - 1)), \gamma\delta)$-DP where $\gamma$ denotes sampling ratio). This yields equation 29.*

*Assuming $\max_{i \in \mathbb{C}} \lambda_i = 1/k$, for $\epsilon_3(\alpha) + \varepsilon_o(\delta) \ll 1$, $\varepsilon_3'(\delta) = \log\left(1 + \frac{k}{n}(e^{\epsilon_3(\alpha) + \varepsilon_o(\delta)} - 1)\right)$ is approximated by $\log\left(1 + \frac{k}{n}(\epsilon_3(\alpha) + \varepsilon_o(\delta))\right)$. Since the log function is a monotone increasing function and $n$ is fixed, $k \cdot (\epsilon_3(\alpha) + \varepsilon_o(\delta))$ should be minimized for the minimum $\varepsilon_3'(\delta)$. Regarding $k \cdot (\epsilon_3(\alpha) + \varepsilon_o(\delta))$, since it is a convex function for $k > 0$, we can find $k_3^*$ which becomes 0 when differentiated. $k_2^*$ can also be calculated in a similar manner. This completes the proof.* ∎

**Revisiting Privacy-Accuracy Trade-Off ($k < n$).** First, privacy-accuracy trade-off between DP-MixSL and DP-CutMixSL occurs in corollary 1 as in theorem 1 where $n = k$. The existence of an optimal

Table 1: Parameters for DP measurement.

| Parameter | Annotation | Value |
|---|---|---|
| Number of clients | $n$ | 10 |
| Dimension of smashed data | $D_s$ | 10 |
| Dimension of label | $D_y = L$ | 2 |
| Pixel-wise upper bound of smashed data | $\Delta$ | 0.15 |
| Mixing ratio | $\lambda_i \ \forall i$ | $1/k$ (uniform) |
| RDP parameter | $\alpha$ | 2 |
| DP parameter | $\delta$ | 0.0002 |
| Noise variance | $\sigma_s^2, \sigma_y^2$ | $\{8/255, 16/255, 32/255\}$ |

mixing group size is rooted in subsampling. In the existing Mixup or Random CutMix, the privacy guarantee improves when the number of samples increases, but counterintuitively, with subsampling, the randomness of which client among all clients a specific sample belongs to decreases, resulting in a trade-off.

**Limitations on DP Analysis.**     There are 2 major limitations on our DP analysis. Firstly, in existing DP analysis, it fundamentally measures how sensitively the output changes compared to the input, and at this time, the output is in-practice bounded (for example, classification). However, since SL inherently lacks in quantifying the change of smashed data versus input, we indirectly analyze the privacy guarantee of smashed data versus output. Instead, we experimentally measure the robustness against the reconstruction attack in section 5 to ensure privacy guarantee between input and smashed data.

Secondly, DP analysis cannot differentiate between Random CutMix and Vanilla CutMix because it focuses on the quantity rather than the randomness or pattern of the mechanism. Through the robustness of the reconstruction attack in section 5 mentioned above, we bypass the privacy guarantee between CutMix, which is theoretically indistinguishable.

## 5    Numerical Evaluation

While section 4 theoretically demonstrates the privacy guarantee of DP-CutMixSL, this section experimentally analyzes its privacy guarantee and accuracy compared to PSL, SFL Thapa et al. (2020b), and etc. For the experiment, we use the CIFAR-10 dataset Krizhevsky (2009) with a batch size of 128, and table 3 additionally uses the Fashion-MNIST dataset Xiao et al. (2017). As an optimizer, Adam with decoupled weight decay (AdamW) Loshchilov and Hutter (2017) is used with a learning rate of 0.001, and a total of 10 clients each have 5,000 images. Except for table 3, we use the ViT-tiny Touvron et al. (2020) model, and the entire model is split so that the client and server each have an embedding layer and a transformer. Other parameters for DP measurement are in table 1 and a list of notations is available in appendix E.

### A    Privacy Analysis: Robustness Against Membership Inference/Reconstruction/Label Inference Attacks

To measure the robustness of DP-CutMixSL against privacy attacks, we first consider the following three types of privacy attacks: membership inference attack, reconstruction attack, and label inference attack. Among them, membership inference attack and reconstruction attack are privacy attacks that occur in the FP process of DP-CutMixSL and label inference attack in its BP process, respectively. Specifically, an honest but curious server in a membership inference attack attempts to determine whether a particular client's data is used for training, via the uploaded DP-CutMix smashed data $\tilde{\mathbf{s}}_{\{i,i'\}}$ and label $\tilde{\mathbf{y}}_{\{i,i'\}}$ generated by the mixer in equation 5. Similarly, in a reconstruction attack, the server aims to restore the input data of $i$-th or $i'$-th client's input data using the auxiliary network through the uploaded DP-CutMix smashed data. Furthermore, in the label inference attack, a client tries to infer the label of input data used by another client through the cut-layer gradient. For example, assuming that the cut-layer gradient $\nabla_{\tilde{\mathbf{s}}_{\{i,i'\}}} \tilde{L}_{\{i,i'\}}$ in equation 8 is sent to clients $i$ and $i'$, the $i'$-th client can try to infer the label of the $i$-th client by performing a classification with the $i$-th client's gradient $\nabla_{\tilde{\mathbf{s}}_{\{i,i'\}}} (\mathbf{M}_i \odot \tilde{L}_{\{i,i'\}})$ as input, which is included in the entire cut-layer gradient.

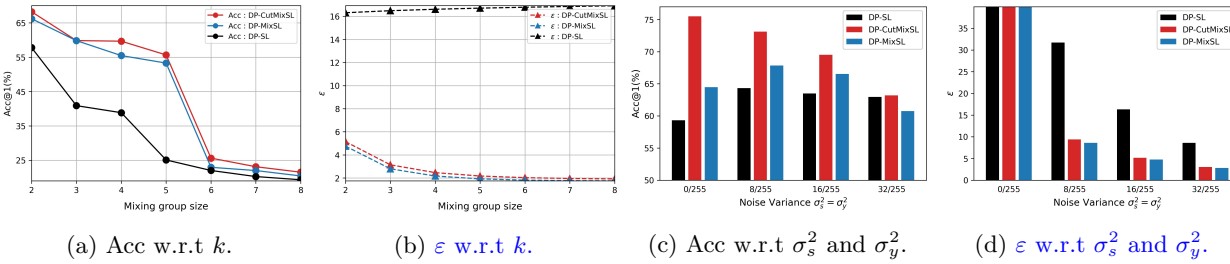

(a) Acc w.r.t $k$.      (b) $\varepsilon$ w.r.t $k$.      (c) Acc w.r.t $\sigma_s^2$ and $\sigma_y^2$.      (d) $\varepsilon$ w.r.t $\sigma_s^2$ and $\sigma_y^2$.

Figure 5: Top-1 accuracy and $\varepsilon$ under the CIFAR-10 dataset: (a) accuracy of DP-CutMixSL and DP-MixSL according to the mixing group size; (b) $\varepsilon$ of DP-CutMixSL and DP-MixSL according to the mixing group size; (c) accuracy of DP-CutMixSL, DP-SL, and DP-MixSL according to noise variance; (d) $\varepsilon$ of DP-CutMixSL, DP-SL, and DP-MixSL according to noise variance.

**Privacy Against Membership Inference Attacks.** Here we assume the worst case that the server knows the smashed data of all client. Note that under this assumption, the DP analysis in section 4 can provide a theoretical measure of the privacy guarantee against membership inference attacks. In addition, hereafter, we use an environment with 64 patches of size $2 \times 2$ and train for 600 epochs for the accuracy measurement.

First, figure 5a and figure 5b shows the effect of mixing group size on accuracy and DP guarantee ($\varepsilon$) of DP-CutMixSL, DP-MixSL, and DP-SL. The first thing to note is the strong privacy guarantee of DP-CutMixSL compared to DP-SL. Factors that improve the DP guarantee of DP-CutMixSL include 1) noise injected by Gaussian mechanism and 2) DP guarantee amplification through Random CutMix, and such gap in privacy guarantee between DP-SL and DP-CutMixSL shows that the latter is more critical to privacy.

Compared to DP-MixSL, regardless of the mixing group size, DP-CutMixSL has higher accuracy compromising a negligible privacy guarantee. Both accuracy and $\varepsilon$ of DP-CutMixSL and DP-MixSL decrease as the mixing group size increases, while those of DP-SL tend to be reversed. This is because in the trade-off of privacy guarantee between subsampling and CutMix or Mixup, the privacy guarantee gain of CutMix or Mixup as $k$ increased is greater than the loss of privacy guarantee due to subsampling, resulting in a "Hiding in the crowd" effect Jeong et al. (2020). It can also be explained by how large the optimal mixing group size is (convex function with respect to $k$), and large $k_2^*$ as well as $k_3^*$ for given parameters validate it ($k_2^* = 28.55$, $k_3^* = 27.07$). On the other hand, DP-SL lacks CutMix or Mixup, so a small $k$ leads to a strong privacy guarantee due to subsampling. Regarding the tendency of accuracy, as shown in figure 10b, the decrease in accuracy for mixing group size is explainable by large-scale noise that is applied to a larger fraction of the image. Furthermore, both DP-CutMixSL and DP-MixSL maintain accuracy until the mixing group size is 5 while improving privacy guarantee.

Figure 5c and figure 5d show accuracy and $\varepsilon$ according to noise variance, respectively. Compared to DP-SL and DP-MixSL, DP-CutMixSL achieves the highest accuracy for all noise variances, while $\varepsilon$ of DP-MixSL is slightly lower than that of DP-CutMixSL. Hereafter, we measure performance on noiseless environments and rename the techniques accordingly, DP-CutMixSL as CutMixSL, DP-MixSL as PSL w. Mixup, etc.

**Privacy Against Reconstruction Attacks.** Table 2 shows the reconstruction loss of SL-based methods according to various hyperparameters. For reconstruction attack, an auxiliary network is used that takes smashed data as input and produces *restored data* through two convolutional layers and interpolate as output. Then, the auxiliary network is trained by minimizing the mean-squared-error (MSE) loss between the restored data and the input data, to generated restored data with high mutual information between the input data. Regarding the hyperparameters, we adjust the dataset size for training the auxiliary network, mask distribution, and mixing group size. Also, examples of restored data generated through the auxiliary network are in appendix C.

| | Training Dataset (10%) | | | | Training Dataset (100%) | | | |
| --- | --- | --- | --- | --- | --- | --- | --- | --- |
| | Mask Distribution ($\alpha_M$) | | | | Mask Distribution ($\alpha_M$) | | | |
| | 2 | | 6 | | 2 | | 6 | |
| mixing group size ($k$) | 2 | 4 | 2 | 4 | 2 | 4 | 2 | 4 |
| PSL | 0.403 | 0.425 | 0.326 | 1.425 | 0.116 | 0.308 | 0.138 | 0.398 |
| PSL w. Mixup | **0.665** | 0.383 | 0.379 | **1.923** | 0.172 | 0.396 | 0.215 | 0.292 |
| PSL w. Vanilla CutMix | 0.382 | 0.426 | 0.429 | 1.316 | 0.180 | **0.403** | 0.219 | 0.417 |
| CutMixSL (proposed) | 0.425 | **0.466** | **0.441** | 1.561 | **0.187** | 0.312 | **0.221** | **0.435** |

Table 2: Reconstruction loss (MSE) of SL-based techniques according to mixing group size, train dataset size, and mask distribution.

First, with respect to the overall tendency for hyperparameters, the larger the training dataset size, the smaller the mask distribution, and the smaller the mixing group size, the more advantageous the auxiliary network is to learn the restored data, leading to a small reconstruction loss.

When comparing SL-based techniques, the reconstruction loss of the proposed CutMixSL is the largest, in other words, CutMixSL outperforms in terms of privacy guarantee for reconstruction attack in most cases, followed by PSL w. Mixup. In particular, when comparing CutMixSL (Random CutMix) and PSL w. vanilla CutMix, the robustness of CutMixSL is superior for most hyperparameter settings. This is due to the difference in randomness between Vanilla and Random CutMix, which is previously indistinguishable by DP analysis. Since adjacent box-shaped pixels are replaced, Vanilla CutMix has a relatively high correlation between pixels, while the correlation between pixels in a Random CutMix that is randomly replaced patch-wise is bounded in patch units, straightforwardly leading to a strong privacy guarantee.

**Privacy Against Label Inference Attacks.** For label inference attacks, there are white-box attacks in Yang et al. (2022), black-box attacks in Li et al. (2021), and other minor variations. We consider a white-box attack among them, since black-box attacks include the bold assumption that clients know the upper model segment weight of the server. Unlike Li et al. (2021), which is based on Vanilla SL, we consider the following worst case of CutMixSL to enable powerful white-box attacks in parallelized form of SL: 1) the first assumption of weight averaging as in SFL to share the weight of the lower model segment among clients, 2) the second assumption that the gradient in the cut-layer is averaged as in Pal et al. (2021) before broadcasting to clients, 3) the third assumption that label inference leakage occurs between clients within the same mixing group size for CutMixSL with $k = 2$. Also, we refer to CutMixSL as its best case when not with the above assumptions. Then, a honest-but-curious client aims to infer its label by measuring the norm (*norm leak*) or cosine similarity (*cosine leak*) of the averaged cut-layer gradient as well as the gradient propagate to the lower model segment.

For measurement, we compute the *area under the ROC curve (AUC)* over the distribution of the norm or cosine similarity. The ROC curve means the curve of the true positive rate (TPR) and false positive rate (FPR) as the decision boundary moves from $-\infty$ to $\infty$ in a binary classification scenario, and if its base area, AUC, is close to 1, it means that the classification of the two distributions becomes clear under accurate data labeling. Thus, in our scenario, an AUC close to 0.5 implies a high privacy guarantee against label inference attacks. As an experimental setting, we utilize the LeNet-5 model LeCun et al. (2015), where the cut-layer is located after the second convolutional layer, and allocate 1,000 samples each corresponding to the two labels 0 and 4 of MNIST dataset to two clients. As a comparator, we use SFL with cut-layer gradient averaging.

Figure 6 and 7 visualize the distribution of norm and cosine similarity according to the mask distribution of CutMixSL and SFL, respectively. where the orange and blue regions each represent that the labels are positive and negative. We can visually confirm that, as the mask distribution increases, CutMixSL does not change significantly, whereas in SFL, the variance of the distribution increases, making it easier to distinguish. Based on these, the ROC curves for norm leak and cosine leak are shown in figure 8a and 8b.

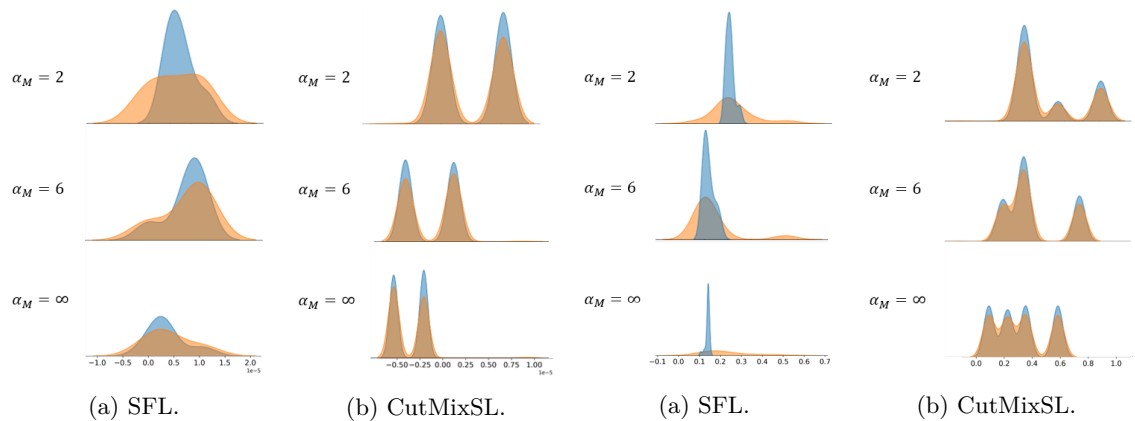

(a) SFL.          (b) CutMixSL.          (a) SFL.          (b) CutMixSL.

Figure 6: Norm distribution of gradients accord-
ing to mask distribution.

Figure 7: Cosine similarity distribution of gradi-
ents according to mask distribution.

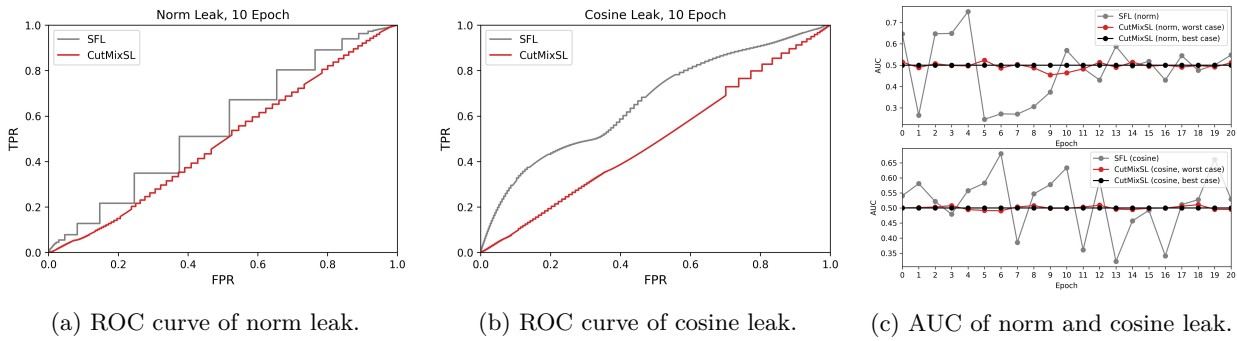

(a) ROC curve of norm leak.          (b) ROC curve of cosine leak.          (c) AUC of norm and cosine leak.

Figure 8: Privacy guarantee measurement for label inference attack of CutMixSL and SFL.

Further, figure 8c measures AUC per epoch of SFL and CutMixSL (its worst case as well as its best case) for norm and cosine leaks. The first thing to note is the strong privacy guarantee for norm and cosine leak of CutMixSL for both cases, maintaining an AUC close to 0.5 for all epochs, thanks to parallelization and Random CutMix's masking effect on gradient of equation 8. The baseline SFL reaches AUCs up to 0.75 and 0.68 for norm and cosine leak, respectively, roughly alleviated by parallelization alone. In addition, regarding the impact of norm and cosine leak according to epoch, the variance of norm leak AUC is larger at the beginning of learning, but becomes weaker as epoch progresses, and instead, the variance of cosine leak AUC becomes large, showing the potential complementary threats of the two privacy attacks.

## B   Performance Analysis: Accuracy & Communication-Efficiency

In this subsection, we evaluate the performance of CutMixSL by dividing it into accuracy and communication-efficiency. In table 3, we additionally measure the accuracy for PiT-tiny Heo et al. (2021a) and VGG-16 Simonyan and Zisserman (2014) except for ViT-tiny. Here, VGG-16 is for CNN in addition to ViT, and PiT is a model between ViT and CNN and is a transformer architecture equipped with a pooling layer. For an extensive comparison of results, we consider PSL with Random Cutout, which is detailed in appendix B, a version of Random CutMix that excludes the Mixup behavior and is also a single-client version of it.

**Accuracy under IID Dataset.**    Table 3 shows the top-1 accuracy on the CIFAR-10 and Fashion-MNIST datasets of various SL-based techniques, including CutMixSL. First, the accuracy of CutMixSL is the highest in all cases except for the case where VGG-16 and CIFAR-10 are used. With VGG-16 and CIFAR-10, PSL w. Mixup achieves the highest accuracy. This is because, as mentioned earlier, CNN focuses on locality when learning spatial information, while ViT focuses on globality. Also, it is consistent with Naseer et al.

Table 3: Top-1 accuracy of methods for various datasets and models.

| Method | Models w/ CIFAR-10 | | | Models w/ Fashion-MNIST | | |
|---|---|---|---|---|---|---|
| | ViT-Tiny | PiT-Tiny | VGG-16 | ViT-Tiny | PiT-Tiny | VGG-16 |
| Standalone | 48.84 | 47.77 | 54.97 | 77.65 | 78.21 | 80.12 |
| PSL | 57.21 | 52.28 | 62.62 | 85.68 | 82.35 | 84.39 |
| SFL | 67.88 | 55.63 | 63.98 | 89.17 | 84.27 | 87.34 |
| PSL w. Mixup | 69.23 | 64.89 | **68.20** | 88.21 | 87.62 | 88.53 |
| PSL w. Random Cutout | 53.86 | 50.28 | 56.65 | 88.46 | 86.48 | 88.17 |
| PSL w. Vanilla CutMix | 71.78 | 58.21 | 33.50 | 87.86 | 86.31 | 89.01 |
| CutMixSL (proposed) | **73.77** | **71.26** | 67.53 | **89.75** | **89.25** | **89.45** |

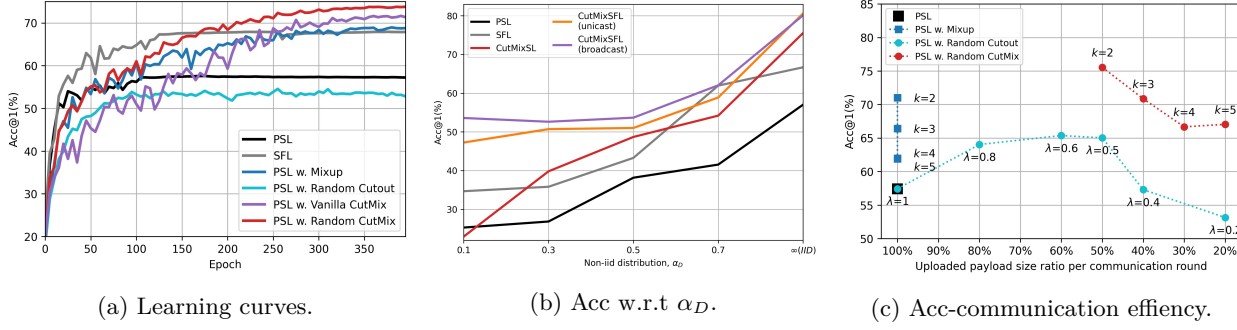

(a) Learning curves.   (b) Acc w.r.t $\alpha_D$.   (c) Acc-communication effiency.

Figure 9: Performance measurements of SL-based techniques: (a) top-1 accuracy over epoch; (b) top-1 accuracy according to non-IIDness. (c) top-1 accuracy and communication payload size for hyperparameters.

(2021), indicating that ViT has robustness of accuracy against patch drop or image shuffling compared to CNN. For that reason, CNN and ViT are better suited for superposition type regularization (i.e., Mixup) and masking type regularization (i.e., Cutout, CutMix), respectively Harris et al. (2020). Compared to PSL w. Vanilla CutMix, CutMixSL is superior in terms of accuracy, which proves the efficiency of the patch-wise designed regularizer in ViT. From the dropout Srivastava et al. (2014) perspective, it can be seen that the more randomized CutMixSL leads to higher accuracy, as in the previous comparison between Vanilla and Random Cutout of figure 11. Furthermore, as in appendix D, Random CutMix achieves the highest accuracy even when applied at the input layer. Straightforwardly, Random CutMix applied to the input layer, however, is more vulnerable to data privacy leakage than that applied to the cut-layer, resulting in an *accuracy-privacy trade-off*. Lastly, figure 9a contains the learning curves of the corresponding techniques on the CIFAR-10 dataset, showing the superiority of CutMixSL in terms of convergence speed as well as accuracy.

**Accuracy under non-IID Dataset.** Figure 9b shows the performance of CutMixSL according to the non-IIDness of the dataset $\alpha_D$. CutMixSL achieves higher accuracy than PSL or SFL, but as non-IIDness becomes more severe (i.e. small $\alpha_D$), the performance drop is greater. To compensate for this, we design *CutMixSFL*, which introduces lower model aggregation as in Thapa et al. (2020b) and SplitAvg as in Pal et al. (2021). As a result, CutMixSFL maintains high accuracy even at low $\alpha_D$ and records the highest accuracy even under the IID condition with $\alpha_D = 9,999$, while compromising the privacy guarantee as discussed in robustness against label inference attack. This is because the server and client can overcome the biased BP flow through the weight and gradient averaging, which will be discussed in detail in future work.

Another interesting point is that in CutMixSFL, thanks to the help of SplitAvg, the averaged gradient can be sent through broadcasting during BP. For example, as in equation 8, in the existing CutMixSL or CutMixSFL without SplitAvg, the mixer unicasts the $i$-th masked gradient $\nabla_{\tilde{\mathbf{s}}_{\{i,i'\}}}(\mathbf{M}_i \odot \tilde{L}_{\{i,i'\}})$ to the $i$-th client, whereas in CutMixSFL with SplitAvg, the server directly broadcasts the averaged gradient to $i(i')$-th client, enabling full band utilization.

**Communication Efficiency.** Figure 9c shows the corresponding accuracy for the uploaded payload size of various SL-based methods. First, we confirm that the increase in communication efficiency of Random CutMix or Random Cutout as $k$ or $\lambda$ increases. Meanwhile, Mixup lacks communication efficiency gains due to its inherent superposition of the entire image. Also, when $k$ increases, the accuracy of Random CutMix decreases, resulting in an *accuracy-communication efficiency trade-off* of Random CutMix.

## 6 Conclusion

In this study, we designed DP-CutMixSL with the goal of developing a privacy preserving distributed ML algorithm for ViT. Thanks to the randomness and masking effect of Random CutMix, we theoretically and experimentally demonstrated that the proposed DP-CutMixSL has robustness against three types of privacy attacks, while not compromising accuracy or communication-efficiency. Although DP guarantee for smashed data was theoretically derived in FP, but our study lacks it in BP. Combined with GradPerturb in Yang et al. (2022), exploring the DP guarantee at BP could be an interesting topic for future work.

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

# Appendices

## A  Observations on Random CutMix

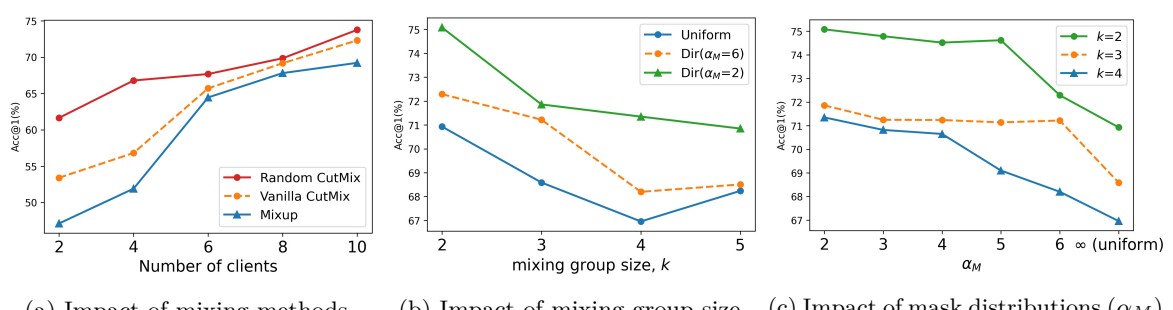

(a) Impact of mixing methods.  (b) Impact of mixing group size. (c) Impact of mask distributions ($\alpha_M$).

Figure 10: Top-1 accuracy of multi-client scenario: (a) accuracy of Random CutMix, Vanilla CutMix, and Mixup w.r.t the number of clients; (b) accuracy of Random CutMix w.r.t the mixing group size; (c) accuracy of Random CutMix w.r.t the mask distribution $\alpha_M$.

In this subsection, the design elements of Random CutMix are explored, along with its optimal hyperparameter settings, especially with respect to its accuracy.

**Random CutMix vs. Vanilla CutMix and Mixup.**  While the proposed Random CutMix is a patch-wise partial regularization scheme rooted in masking, a Mixup Zhang et al. (2017); Verma et al. (2019) that superpositions the full image can be considered as an alternative. When Mixup is applied instead of Random CutMix, equation 5 is replaced with the following formula: $\hat{\mathbf{s}}_{\{i,i'\}} = \lambda_i \cdot \mathbf{s}_i + \lambda_{i'} \cdot \mathbf{s}_{i'}$, $\hat{\mathbf{y}}_{\{i,i'\}} = \lambda_i \cdot \mathbf{y}_i + \lambda_{i'} \cdot \mathbf{y}_{i'}$. Another alternative is Vanilla CutMix, a masking type regularization equal to Random CutMix. In this case, equation 5 remains the same, but while Random CutMix randomly replaces patches, Vanilla CutMix replaces internal pixels based on the box-shape region as in Vanilla Cutout. Figure 10a shows the comparison of accuracy between these regularization schemes according to the number of clients $n$. The top-1 accuracy is high in the order of Random CutMix, Vanilla CutMix, and Mixup regardless of the number of clients, showing superiority of Random CutMix in ViT. Also, in all three regularization schemes, accuracy increases as the number of clients increases, that is, scalability is guaranteed up to 10 clients.

**Impacts of Mixing Group Size and Mask Distributions.**  In figure 10b, the accuracy of the Random CutMix as the mixing group size varies is shown for mask distribution $\alpha_M$. Note here that the infinite divergence of $\alpha_M$ implies that the mixing ratio follows a uniform distribution. Without cases where $k$ is 4 with $\alpha_M$ of 2 or 6, the top-1 accuracy tends to be inversely proportional to the mixing group size, especially its decline is greatest when $k$ changes from 2 to 3. Interpreting this from the perspective of each client, as the mixing group size $k$ increases, large noise that may lead to performance degradation is applied in the remaining areas except for $\frac{1}{k}$ of the entire image, under the assumption of a uniform mixing ratio. For similar reasons regarding distortion level, figure 10c includes a tendency for accuracy to decrease as $\alpha_M$ increases.

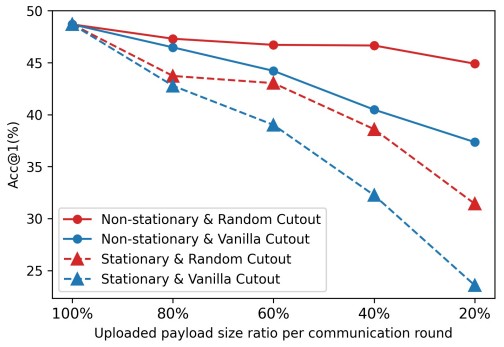

(a) Top-1 Accuracy according to communication cost.

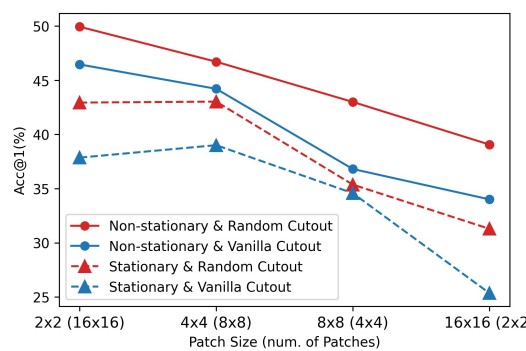

(b) Top-1 accuracy according to patch size.

Figure 11: Accuracy of a single client using Vanilla or Random Cutout whether a mask is stationary or not.

## B  Random Cutout

This subsection introduces Cutout-based version of Random CutMix, which is also equivalent to its single-client case. Starting with the Vanilla Cutout, this subsection sequentially covers the client-side behavior, design elements, and characteristics of the Random Cutout and provides some insights from them..

**Vanilla Cutout.**     After the patch embedding process, the $i$-th client of Vanilla Cutout DeVries and Taylor (2017) masks out the smashed data as follows:

$$\mathbf{s}'_i = \mathbf{M}_i \odot \mathbf{s}_i, \tag{33}$$

where $\mathbf{M}_i$ is a binary matrix in which the elements belonging to the inside are 0 and the rest are 1 for a random bounding box denoted by $\mathbf{B} = (r_x, r_y, r_w, r_h)$, $\odot$ is an pixel-wise multiplication operator.

Then, $\mathbf{s}'_i$ is sent to the server, followed by a server-side FP and BP process of subsection A, completing a single communication round of Vanilla Cutout. At this time, the loss generated by the server is the same as in equation 2, where $f_i(\mathbf{x}_i)$ is replaced with $\mathbf{s}'_i$.

**Random Cutout.**     In Random Cutout, $\mathbf{M}_i$ in equation 33 is a patch-wise on-off matrix and is denoted as $\mathbf{M}_i = [\mathbf{M}_{i,0}, \mathbf{M}_{i,1}, \cdots, \mathbf{M}_{i,N}]$, consisting of $\mathbf{M}_{i,j}$ for $j \in \{0, 1, \cdots, N\}$ that is mapped with the corresponding embedded patch $\mathbf{s}_{i,j}$ and determines whether to drop it.

Figure 11 shows that Random Cutout outperforms Vanilla Cutout in terms of top-1 accuracy for all cases. This is consistent with our reasoning that patch-wise regularization would be more suitable for ViT, where all operations are patch-wise. Also, from the point of view of dropout, which is often pointed out as the main reason for the high performance of Vanilla Cutout, it can be similarly inferred that the higher performance of the Random Cutout is rooted in a more dropout-like property (i.e. higher randomness) than Vanilla Cutout. In addition, the Random Cutout includes design elements, as elaborated below.

**Impact of Mask Stationarity.**     For the Random Cutout as well as the Vanilla Cutout, we can divide the cases according to how often $\mathbf{M}_i$ is updated, so in figure 11 we compare the two extreme cases, updated every communication round (non-stationary) or not updated (stationary). Comparing them, the Random or Vanilla Cutout always has high accuracy with the non-stationary mask, which is advantageous for capturing more generalized global spatial information, leading to avoidance of overfitting. In addition, figure 11a shows the accuracy with respect to the communication cost, in other words, the payload size. Here, the uploaded payload size is proportional to the number of 1s among the elements of $\mathbf{M}_i$.For example, the uploaded payload size ratio of 100% means that all elements of $\mathbf{M}_i$ are 1 and the Cutout smashed data is equal to the smashed data. Figure 11a indicates that the top-1 accuracy of both Cutouts decreases as the ratio of uploaded payload size increases, showing the accuracy and communication cost trade-off. Among them, Random Cutout with non-stationary mask has the smallest drop in accuracy.

**Impact of Patch Size.** Regarding patch size, figure 11b demonstrates that the smaller the patch size, the higher the overall accuracy tends to be. Due to the characteristics of the transformer as well as ViT, which learns the relative spatial relationship between patches through positional embedding, small patch size allows for granular and generalized classification.

## C Visualization of Additional Images

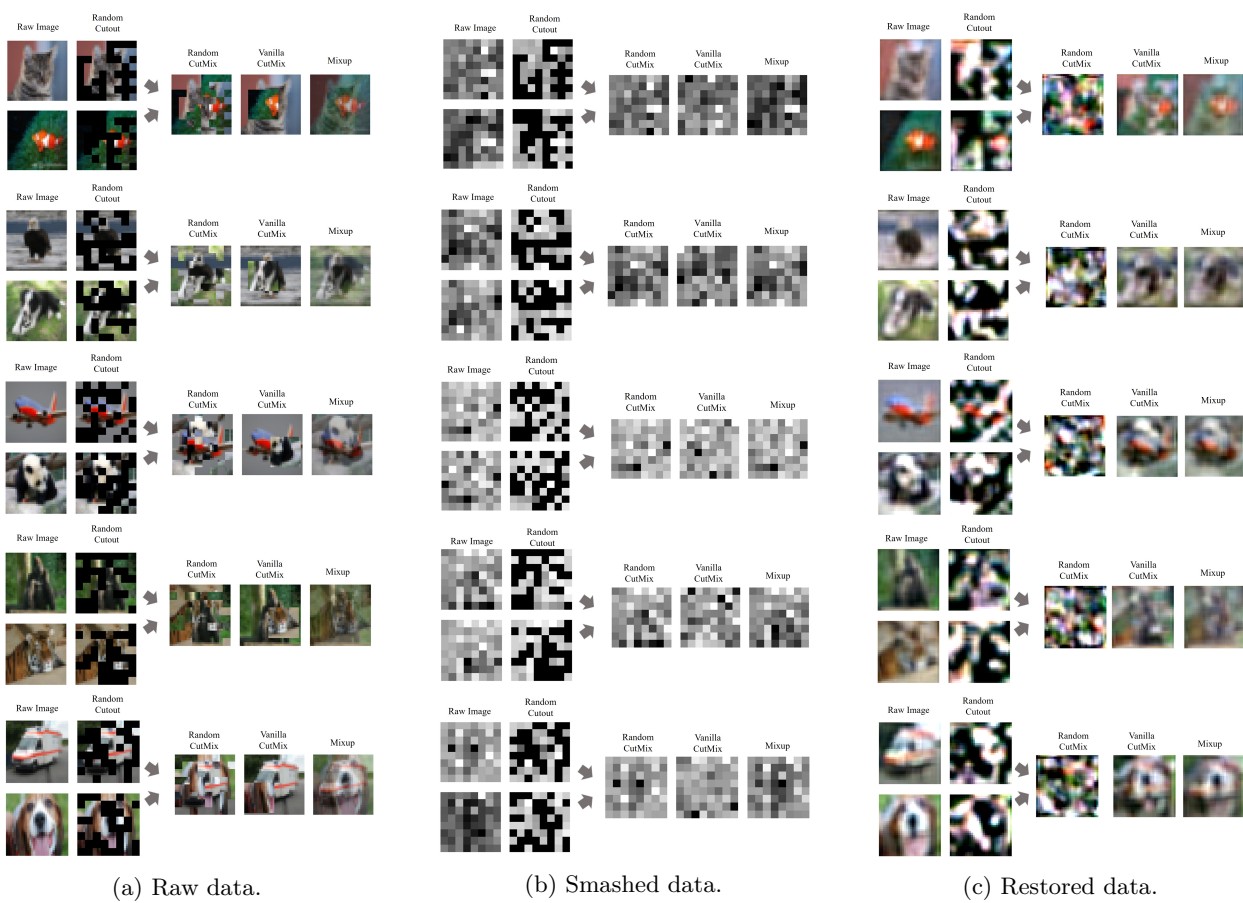

(a) Raw data.     (b) Smashed data.     (c) Restored data.

Figure 12: Additional example images for various regularization techniques.

## D Accuracy comparison of regularizations applied to the input layer

Table 4: Top-1 accuracy of SL-based methods for various datasets and models.

| Method | Models w/ CIFAR-10 | | | Models w/ Fashion-MNIST | | |
|---|---|---|---|---|---|---|
| | ViT-Tiny | PiT-Tiny | VGG-16 | ViT-Tiny | PiT-Tiny | VGG-16 |
| PSL w. Mixup | 74.36 | 37.21 | 66.08 | 89.86 | 87.62 | 89.70 |
| PSL w. Random Cutout | 22.03 | 45.19 | 66.32 | 88.65 | 88.51 | 89.62 |
| PSL w. Vanilla CutMix | 73.02 | 33.54 | 47.69 | 88.72 | 88.37 | 90.02 |
| CutMixSL (proposed) | **75.06** | **53.93** | **67.43** | **89.91** | **89.53** | **90.32** |

## E   List of Notations

Table 5: List of Notations.

| Notation | Meaning |
|---|---|
| $\mathbb{C}$ | a set of clients |
| $n$ | # of clients |
| $\mathbb{D}_i$ | dataset of the $i$-th client |
| $\mathbf{x}_i$ | sample of the $i$-th client |
| $\mathbf{y}_i$ | label of the $i$-th client |
| $\mathbf{w}_{c,i}$ | lower model segment of the $i$-th client |
| $\mathbf{w}_s$ | upper model segment of the server |
| $P$ | patch size |
| $N$ | # of patches |
| $H, W, C$ | height, width, channels of image |
| $\mathbf{s}_i$ | smashed data of the $i$-th client |
| $D_s$ | dimension of $\mathbf{s}_i$ |
| $D_y$ | dimension of $\mathbf{y}_i$ |
| $\mathbb{B}_i$ | batch of the $i$-th client |
| $b$ | batch size |
| $\eta$ | learning rate |
| $\mathbf{M}_i$ | binary on-off matrix for $\mathbf{s}_i$ |
| $k$ | mixing group size |
| $\mathbb{C}_k$ | a set of mixing groups of size $k$ |
| $\lambda_i$ | mixing ratio of $i$-th client |
| $\alpha_M$ | mask distribution |
| $\alpha_D$ | dataset distribution |
| $\tilde{\mathbf{s}}_{\{i,i'\}}$ | $i$-th DP-CutMix smashed data |
| $n_{s,i}$ | the $i$-th zero-mean Gaussian matrix with variance $\sigma_s^2$ and size $D_s$ |
| $n_{y,i}$ | the $i$-th zero-mean Gaussian matrix with variance $\sigma_y^2$ and size $D_y$ |
| $\alpha$ | RDP parameter |
| $\delta$ | DP parameter |
| $\Delta$ | pixel-wise upper bound |

