# OpenReview forum: "Patch-Wise Random and Noisy CutMix for Privacy-Preserving Split Learning with Vision Transformer"
_TMLR — Rejected by TMLR_

### Review · Reviewer_ncWh · 2023-03-09

**Summary Of Contributions:**

The paper proposes a patch-wise version of CutMix data processing technique for split learning of vision transformer with differential privacy guarantee. The key idea is to add patch-wise mask and noise and perform group-wise mixup at a trusted third party before sending data to a honest-but-curious server.

**Audience:**

Yes

**Broader Impact Concerns:**

No concerns.

**Claims And Evidence:**

Yes

**Requested Changes:**

1. In figure 7, how is the privacy epsilon calculated? Is it for per iteration privacy guarantee or the overall composited epsilon of all iterations required to training the model? How many iterations the model used? It would be better to highlight these details for readers to clearly understand the configuration.


**Strengths And Weaknesses:**

Strengths:
1. The proposes approach outperforms the baseline which is directly adding noise on data in split learning, the differential privacy guarantee is also analyzed for the proposed approach. Experiments on inference/reconstruction attacks are also done to show the effectiveness of the proposed algorithms.
2. The paper include a detailed reasoning of the proposed approach and discussion on existing literature. It is well-organized and easy to follow.

Weaknesses:
1. The proposed paradigm requires a trusted third party which is sometimes hard to satisfy in practice.
2. It is hard to quantify how practical or meaningful is the privacy guarantee when training models. I am assuming the individual parameters for each client will be updated periodically and each time it is updated, the patches will need to be updated and it will be considered as a new query to data. Because training models requires so many iterations and the composition of private queries will make overall privacy guarantee very bad. Otherwise, if a lot of noise is added to guarantee privacy, the learned model performance could be bad. Please correct me if I am wrong about the configuration.

---

> ### Author Response · Authors · 2023-04-13
> **Response to Reviewer ncWh**
>
> We are thankful to the reviewer for your constructive feedback regarding our manuscript.
>
> 1. Regarding mixer: As we mentioned before, the mixer can be 1) homomorphic encryption on the server, 2) an analog channel, and 3) a trusted third party such as a mobile network operator. As you pointed out, the third option can be hard to satisfy in practice, so we have revised the manuscript to only mention 1) and 2).
>
> 2. Regarding DP measurement: When measuring DP, we did not measure the accumulated privacy over iterations, but rather measured the DP bound of each mechanism for a single iteration, as is commonly done. In a scenario where the pixel-wise upper bound (denoted as $\Delta$) of the smashed data changes over time, it might be meaningful to calculate the cumulative privacy bound as you suggest. However, in this study we assumed it to be a fixed value, so we defer it to future work. We also have added the number of epochs for the accuracy measure in the revised manuscript for the detailed simulation settings you requested.

---

### Review · Reviewer_ULeo · 2023-03-12

**Summary Of Contributions:**

This paper discusses split learning vision transformers in federated learning. Several components are discussed, e.g., (1) random cutout – patch-wise mask instead of pixel wise compared to vanilla cutout (2) a mixer uses mixup on top of random cutout (3) adding noise for differential privacy. It seems that DP-CutMixSL is proposed by combining the three components.


**Audience:**

Yes

**Claims And Evidence:**

No

**Requested Changes:**

See my comment above.

**Strengths And Weaknesses:**

This draft seems to be unnecessarily long and introduces a lot of (unnecessary) concepts. Even if I spent quite some time on this draft – at least doubled the usual reviewing time for a single paper, I am still not confident about my assessment. The current paper also looks like a combination of several shorter papers. I would strongly encourage the authors to improve the flow of the paper, reduce the notation, try to use figures to replace equations, provide an overview of the method before diving into details, and most importantly, discuss motivation for why the method is proposed.

Why is split learning necessary? Why operate on smashed data in splitting learning, instead of masking/cutout and noising the input images? Figure 5 seems to be related, but I am not sure I am convinced.

Many unclear component of the DP part:
How is DP defined? Specifically, what are the neighboring datasets, and what is the random process for DP-SL, DP-MixSL and DP-CutMixSL? For D in definition 1, is it assuming each client only has one data point, and subscript i is used to represent both data and client?

How are sensitivity computed?

In Theorem 1, why can the same noise amount (\sigma_s, \sigma_y) be assumed to compare various methods? I cannot understand the intuition why one method is better than the other.

Could the authors discuss what exactly are done for privacy attacks: membership inference attack, reconstruction attack, and label inference attack?

In table 1, \delta~0.5 seems to be very large, as \delta should be smaller than the inverse of the dataset size.

---

> ### Author Response · Authors · 2023-04-13
> **Response to Reviewer ULeo**
>
> We are thankful to the reviewer for your constructive feedback regarding our manuscript.
>
> 1. Regarding readability: Based on your suggestions, we have revised the overall structure of the paper. First, to reduce the amount of unnecessary concepts and notations in the main text, we have moved Random Cutout and some properties of Random CutMix in Section 3 to the appendix, and added figure 3 and revised the overall wording of Section 3.2 to clarify the proposed DP-CutMixSL. We have also made the motivation of the paper in Section 3.1 clearer. In addition, to make Section 4 easier to understand, we have modified the presentation flow in Section 4 by sequentially measuring their behavior and RDP bounds from the baseline DP-SL to DP-CutMixSL, and comparing the results afterwards.
>
> 2. Regarding split learning: Considering the constraints on device-side memory size and the computationally expensive nature of ViT training, split learning can be an alternative soluition to the widely-used federated learning, which can also give data diversity gain to ViT with low inductive bias. The approach you mention, regularization on raw data instead of smashed data, has a generally larger accuracy gain as also shown in appendix D of the manuscript. Howerver, its privacy leakage is greater than that of smashed data, which can be particularly fatal in ViT, where the privacy leakage is comparatively larger than conventional CNN, as pointed out in the manuscript. To avoid further confusion, we have added a sentence about the accuracy-privacy trade-off of applying regularization in the input layer versus the cut-layer in conjunction with Tables 3 and 4.
>
> 3. Regarding DP measurement: For all mechanisms (DP-SL to DP-CutMixSL), we applied the Gaussian DP mechanism. What we mean by neighboring datasets is datasets with different single elements and the same remaining elements, and DP-SL, DP-MixSL, and DP-CutMixSL include random processes corresponding to adding Gaussian noise to smashed data, Mixup on smashed data followed by Gaussian noise, and Random CutMix to smashed data followed by Gaussian noise, respectively, which are more clearly explained in the revised manuscript. Also, as you pointed out, in Section 4, subscript $i$ is used to represent client and its data point together, and we have added an assumption about this to Section 4 of revised manuscript.
>
> 4. Regarding sensitivity: Since we assume a Gaussian DP mechanism, the sensitivity $s$ is calculated as follows: $s=\sigma\varepsilon/\sqrt{2\log(1.25/\delta)}$. Based on our parameter settings and the results in fig 5(b), the sensitivity $s=0.03$ for $\varepsilon=2$ and sensitivity $s=0.24$ for $\varepsilon=16$, respectively.
>
> 5. Regarding noise amount: Taking the case of DP-CutMixSL as an example, its DP bound is determined by 1) noise injection following Gaussian DP mechanism, and 2) DP amplification due to Random CutMix. Under these circumstances, we set the same noise variance in order to fairly compare the DP amplification effect of DP-CutMixSL or DP-MixSL compared to DP-SL, including only 1). We believe that the sequential explanation of Section 4 in the revised manuscript will reduce confusion about this.
>
> 6. Regarding privacy attacks: The membership inference attack is carried out by a honest but curious server who uses the DP-CutMix smashed data $\tilde{\mathbf{s}}{{i,i'}}$ and label $\tilde{\mathbf{y}}{{i,i'}}$ generated by the mixer in equation (5) to determine if a specific client's data is used for training. In addition, the reconstruction attack involves the server attempting to restore the input data of either the $i$-th or $i'$-th client using the auxiliary network through the uploaded DP-CutMix smashed data. The label inference attack, on the other hand, is carried out by a client attempting to infer the label of another client's input data using the cut-layer gradient. A detailed description of these privacy attacks mapped with formulas in Section 3 is now available in front of Section 5.1 of the revised manuscript.
>
> 7. Regarding $\delta$: As per your suggestion, in the revised manuscript, we have reprinted the results of figure 5(b) and 5(c) according to $\delta=0.0002$, and have confirmed that the trend after correction was maintained as before.

---

> > ### Comment · Reviewer_ULeo · 2023-04-19
> > **Sensitivity**
> >
> > Thanks for the revision, I will read the updated paper.
> >
> > I want to  comment again as I do not understand the response for sensitivity. Why is sensitivity computed backward from DP guarantees? Sensitivity has to be controlled by the mechanism/algorithm to get DP guarantees. How are s=0.03 and s=0.24 enforced?

---

> > > ### Author Response · Authors · 2023-04-21
> > > **Regarding Sensitivity**
> > >
> > > Thank you for pointing this out.
> > >
> > > For a given pixel-wise upper bound $\Delta$ of smashed data and its dimension $D_s$, the L2-sensitivity of DP-CutMixSL on a sample is $\sqrt{\Delta^2 D_s/n}$ as shown in Equation 22 (assuming $\lambda=1/n$). Also, the L2-sensitivity for label in DP-CutMixSL is $\sqrt{D_y/n^2}$, the same as in DP-MixSL.
> > >
> > > Similarly, the L2-sensitivity of DP-SL and DP-MixSL can be calculated from equations (15) and (18) in the manuscript.
> > >
> > > For $n=10$, a comparison of them with the parameters given in Table 1 is as follows.
> > > mechanism | L2-sensitivity for sample | L2-sensitivity for label |
> > > |-----------|-----------|-----------|
> > > DP-SL | $\sqrt{\Delta^2 D_s}=0.47$ | $\sqrt{D_y}=1.41$ |
> > > DP-MixSL| $\sqrt{\Delta^2 D_s/n^2}=0.05$ | $\sqrt{D_y/n^2}=0.14$ |
> > > DP-CutMixSL| $\sqrt{\Delta^2 D_s/n}=0.15$ | $\sqrt{D_y/n^2}=0.14$ |

---

### Review · Reviewer_jwkb · 2023-03-30

**Summary Of Contributions:**

This paper proposes a way to train vision transformers in "split learning" (lower layers on the client side and upper layers on the server side) in privacy-preserving way. The authors propose to use noisy random cutmix to obfuscate information that goes to the server, and verify that their solution is privacy-preserving by performing membership inference, reconstruction and label inference attacks. The author show that one of their techniques (DPCutMixSL) satisfies differential privacy and as such is theoretically robust against these attacks. The authors experiment on CIFAR-10 and FashionMNIST, showing how various techniques impact the test accuracy of the models and how they leave vulnerabilities to e.g. reconstruction attacks.

**Audience:**

Yes

**Broader Impact Concerns:**

The work tackles privacy-preserving AI which is improving AI ethics.

**Claims And Evidence:**

Yes

**Requested Changes:**

I think this paper needs more ablation studies as it proposes a very complicated pipeline. I would start with a baseline of training one network on one server in a privacy-preserving way and then add each component one-by-one to better see the impact of it.

**Strengths And Weaknesses:**

Strengths
- The paper tackles an interesting and important problem: training vision models in a privacy-preserving way in a distributed client-server setup
- The paper leverages state-of-the-art techniques in vision to apply them in the privacy-preserving setup

Weaknesses
- I find the paper a bit difficult to follow, there are many names like "smashed data" that are not well defined and that we could do without
- In general, the set-up of training the first layers on the client is a bit weird to me: the first layers of a neural network are supposed to be "general purpose" while the last layers are higher-level and depend on the distribution.
- The resulting accuracies of the networks are bad (Figure 3, all accuracies are below 50%)
- The differential privacy part is also a bit weird: it adds a lot of noise as it is "per client" (equation 5), the authors spend some time on the proving that it's DP but it seems quite standard to me (Section 4), and it ends up being used with delta=0.5 (Figure 7) which is much higher than what is recommended (delta=1/N is the standard).
- The privacy-preserving property comes from DP but to the best of my knowledge the cutmix aspect does not improve DP, so why add cutmix to DP?

---

> ### Author Response · Authors · 2023-04-13
> **Response to Reviewer jwkb**
>
> We are thankful to the reviewer for your constructive feedback regarding our manuscript.
>
> 1. Regarding readability: Following your suggestions, we have made changes to the paper's overall structure, especially in Sections 3 and 4. We have moved the Random Cutout in Section 3 to the appendix to reduce the number of definitions as well as notations. Moreover, we have modified the description in Section 4 to start with the baseline DP-SL and add operations one by one to calculate the $\epsilon$.
>
> 2. Regarding cut layer: To best of our knowledge, under a split learning framework, the higher the cut-layer, the lower the accuracy gain when applying regularization to smashed data in general, as well as the gain in terms of memory size on the client side. This can also be intuitively inferred by comparing tables 3 and 4 in the manuscript. Nevertheless, as you suggest, placing the cut-layer higher is expected to improve the privacy guarantee leading to an accuracy-privacy trade-off, and we will leave it for our future research.
>
> 3. Regarding accuracy: As our experiment is based on a model with a relatively small number of parameters, such as ViT-Tiny, and the size of the training dataset only reaches 1/10 of the total, the standalone accuracy is <50% as shown in Table 3. Nevertheless, the proposed DP-CutMixSL successfully reached >70% accuracy (up to 73.8%) in the same environment.
>
> 4. Regarding $\delta$: Based on your suggestion, we have revised the manuscript to reprint the results of Figure 5(b) and 5(c) using a value of $\delta=0.0002$, and have confirmed that there was no difference in the tendency for $\varepsilon$ after modification.
>
> 5. Regarding CutMix: Neither CutMix nor Mixup can be DP mechanisms by themselves, but they can be utilized as DP amplifiers when combined with Gaussian DP mechanisms, of which Mixup has been theoretically proved in [Lee2019Synthesizing]. As explained in Section 4.3 of the revised manuscript, our proposed Random CutMix consists of Random Cutout, Mixup, Gaussian DP mechanism, and among them, Random Cutout can intuitively operate as a DP amplifier by cutting off part of the image, so the Random CutMix is also a DP mechanism. Consequently, Theorem 1 shows its DP bound, improved compared to that of DP-SL but unfortunately inferior to that of DP-MixSL.
>
> [Lee2019Synthesizing] Kangwook Lee, Hoon Kim, Kyungmin Lee, Changho Suh, and Kannan Ramchandran. Synthesizing differentially private datasets using random mixing. In 2019 IEEE International Symposium on Information Theory (ISIT), pages 542–546. IEEE, 2019.

---

### Decision · Action_Editors · 2023-05-12

**Recommendation:** Reject

**Comment:**

Based on both my perspective and the feedback from the reviewers, there are some critical areas where the paper could see significant improvement. Firstly, the overall complexity and length of the paper present challenges to comprehension. There seems to be an overabundance of concepts introduced without clear and concise explanations, which makes it difficult to follow the authors' line of thought. Simplifying the content, reducing the use of heavy notation, and possibly using more figures to replace equations could make the paper more accessible and digestible.

More critically, we share concerns about the approach the authors have taken in evaluating their differential privacy (DP) claims. The method of measuring DP per iteration, rather than cumulatively across the entire training process, deviates from standard practice. This approach makes it challenging for us to validate the effectiveness and performance of the proposed algorithm accurately.

In summary, a more streamlined presentation of the concepts, clearer explanations, and a more conventional evaluation of DP claims could significantly improve the paper. These modifications would make the work more understandable and convincing.

**Audience:**

The topic is of interest to the TMLR audience, but the topic is somewhat narrow.

**Claims And Evidence:**

This paper introduces a privacy-preserving technique for training vision transformers in a split learning setting, utilizing a combination of patch-wise masking (random cutout), mixup, and noise addition (DP-CutMixSL). The technique, which includes obfuscation via noisy random cutmix, offers differential privacy guarantees and is shown to be robust against membership inference, reconstruction, and label inference attacks. The authors conducted experiments on CIFAR-10 and FashionMNIST datasets, highlighting the impact on test accuracy and vulnerability to attacks.